# SOD2G: A Study on a Social-Engineering Organizational Defensive Deception Game Framework through Optimization of Spatiotemporal MTD and Decoy Conflict

**Sang Seo and Dohoon Kim ***

Department of Computer Science, Kyonggi University, Suwon-si 16227, Korea; tjtkd8271@kyonggi.ac.kr
* Correspondence: karmy01@kyonggi.ac.kr

**Abstract:** Existing moving target defense (MTD) and decoy systems are conceptually limited in avoiding and preventing attackers' social-engineering real-time attacks by organization through either structural mutations or induction and isolation only using static traps. To overcome the practical limitations of existing MTD and decoy and to conduct a multi-stage deception decision-making in a real-time attack-defense competition, the current work presents a social-engineering organizational defensive deception game (SOD2G) as a framework, consi dering hierarchical topologies and fingerprint characteristics by organization. The present work proposed and applied deception concepts and zero-sum-based two-player game models as well as attacker and defender decision-making process based on deceivable organizational environments and vulnerability information. They were designed in consideration of limited organizational resources so that they could converge in the positive direction to secure organizational defender dominant share and optimal values of the defender deception formulated by both scenario and attribute. This framework could handle incomplete private information better than existing models and non-sequentially stratified, and also contributed to the configuration of the optimal defender deception strategy. As the experimental results, they could increase the deception efficiency within an organization by about 40% compared to existing models. Also, in the sensitivity analysis, the proposed MTD and decoy yielded improvements of at least 60% and 30% in deception efficiency, respectively, compared to the existing works.

**Keywords:** defensive deception; moving target defense; decoy; social-engineering; game theory

## 1. Introduction

Defensive cyber deception [1,2] is a type of non-cooperative decision-making contaminating technology that involves manipulating the cognitive perspective of potential attackers and deceiving them so that they continuously construct and maintain erroneous post-action strategies. This is carried out by leveraging defender-dominant information asymmetry. Defensive cyber deception is a new concept [3,4] with unique characteristics that distinguish it from other security elements, such as induction, isolation, backtracking, and mutation, while separately having dedicated kill chain processes by application environment and scenario. By introducing defensive deception as such, the attacker-dominant spatial and temporal asymmetry remaining in conventional security (such as access control and intrusion blocking) can be drastically alleviated in existing system architectures. Related deceptive security systems can be also flexibly applied and distributed in multiple operating layers more easily without having to make more substantial changes than in conventional security. It can also serve as a major source of technology, driving the development of the concept of active defense in an organizational operating environment. Its expected benefits include dynamically

detecting and collecting artifacts such as attackers' attack vectors or intrusion paths to trace them back, formulating all immediate detection, proactive evasion, induction, and isolation schemes by advanced persistent threat (APT) type, and maximizing attackers' total attack cost while reducing defenders' entropic time spent to capture infiltration attempts [5].

## 1.1. Background

Defensive cyber deception technology includes MTD [6], Honey-X [7], and Decoy [8,9]. It can be classified according to its operational goals and defensive purposes. For example, based on its mobility properties such as shuffling, diversity, and redundancy, MTD can limit the effectiveness of defender intelligences collected in advance by attackers either periodically or aperiodically, mutate observable attacks and exploration surfaces [10] according to the defender's intention to increase the attacker's uncertainty and confusion, thus blocking serial compositions of attack chains and ultimately resolving the attacker dominant information asymmetry to realize a proactive defense. Meanwhile, Decoy and Honey-X are dynamic sandboxing elements. They can perform induction, isolation, and decoy schemes known to cause attackers to attack false targets that are not protected. These can interact with each other to mislead attackers based on the network or system level depending on the defender's intention. MTD has been combined with decision-making strategies among decision-making entities in areas such as game theory [11–14], MDP (Markov decision process) [15–18], and machine learning [19–22]. It has also been combined with the learning theory for benefit optimization between attacks and defenses to achieve diverse optimized mutation strategies and deception thresholds while attenuating effects of intrusion by cyber kill chain (CKC) [23] stage and surface spaces. And this MTD has also been combined with other elements such as Honey-X and Decoy.

## 1.2. Limitation of Existing Defensive Deception Based on Non-Common and Common Games

However, the existing MTD and Decoy design presented in previous studies has a few limitations. Limitations of the non-game theory-based defensive deception studies are as follows:

- Organizational deception management strategies are not considered: To improve the reliability and practicality of defensive cyber deception in an organization and to optimize the common goal of maximizing security while minimizing the impact on performance, deception modeling dedicated to MTD and Decoy should be constructed and deception management strategies tailored to unique characteristics of each organization should be formulated. This is also related to the seamless connection of legal services enabled by the application of MTD [24,25]. However, the actual operability of this setup is low due to the fact that previous studies have responded by opening separate encrypted closed communication channels and mapping random virtual network information with real information, thus partially mutating the information. In addition, since the generated virtual information is also randomly composed, the attacker's judgment cannot be adjusted according to the intention of the defender. In addition, the artificial cognitive bias is insignificant.
- Having a low flexibility and scalability: The existing naive MTD is only focused on securing proactive defensibility such as through obstructing the formation of attack chains by evading attackers' initial reconnaissance through structural mutations or judging those users who have requested illegal characteristic information (e.g., IP address, port number, service, etc.) at random points in time as illegal and then inducing and isolating them into static traps. In addition, since post-responses by threat were not considered as major actions in the MTD concept, there was also a complete lack of countermeasures against deviations of sophisticated attackers who had identified MTD mutation patterns or fatal vulnerability attack points in the organization's operational architecture.

- Lacking deception strategies against social engineering APT attacks: Quantitative deception management strategies against various social engineering APT attack vectors [26], such as malvertising [27], spear phishing [28], and watering hole [29], should be presented along with optimization plans. That is, the skewness and outliers in the detection and blocking system of an organization with limited available resources should be alleviated. Mutations of MTD related to open-source intelligence (OSINT) [30,31], HUMINT, Decoy management-distribution, dynamics diversification of attack, exploration surfaces, and so on should all be considered.
- Defender deception supplementation and intelligence composition are not systematized: Although many approaches can be used to reduce the skewness or errors in threat intelligences and improve response quality using research models and platforms integrating existing MTD and decoys, they are limited to the test bed configuration based on certain domains. In addition, the conceptualization of active defender deception actions appropriate to the actual organizational environment has also been limited to studies involving a small number of vendors based on the concept of disinformation and adaptive pollution of external OSINT [32].

Limitations of game theory-based deception studies can be summarized as follows:

- Having decision-making conflicts and making it impossible to optimize deceptions: Game theories are used to model dynamic decision-making in uncertain situations under the assumption that each player has a consistent view. However, when decision-making is based only on asymmetric information that can be observed by the player at random points in the same game, an unoptimized deceptive value can be calculated depending on the game strategy and the uncertainty of the equilibrium state [33]. Therefore, a separate concept of view and share considering the presence of uncertainty that can lead to different optimal judgments each time should be added.
- Concept of high dynamics and real-time quality are not considered: Although many different types of disturbances and contingent factors in the network can affect the security state in an organization with rapid changes of the attack-defense environment, previous studies did not accurately characterized real-time changes in the performance of the actual network attack-defense process or consider the noise that can potentially affect the rewards by episode since they sequentially modeled the relationship between defenders and attackers mainly based on simple leaders and followers. In addition, when players are taking actions intended to achieve their goals, it is impossible to dynamically calculate suboptimal deception strategies against incompletely perceived enemies due to the sequential consumption of episodes for equilibrium [34].
- Lacking organizational unique characteristics and operating principle: Since previous game theory studies have defined attack-defense targets, equilibrium states, weights, and state-transition probabilities in either a static manner or a weak dynamic manner based on CVE (common vulnerability-ties and exploits) and CVSS (common vulnerability scoring systems) in NVD (national vulnerability database) when constructing scenarios and sequences, they are efficient in comparing and verifying the efficiency and utility of defender deception based on vulnerable points of contact and attack vectors. They can be used to calculate the effectiveness of a dedicated system reflecting some characteristics of a simulated domain (e.g., multi-tenant cloud network [35], vehicle-type internal/external communication networks [36,37]) depending on the definition of the topology structure. However, measures for applying the concept of defensive cyber deception to the actual organizational environment have not been officially reported yet. They do not have the dedication of the concept of deception according to organizational unique characteristics, operating principles, or possible attack-defense patterns, etc.

This study of social engineering defensive organizational deception games aims to alleviate practical limitations and theoretical scalability issues of non-game theory- and game theory-based defensive cyber deception studies and to practically model the decision-making process between attackers and defenders in an organizational environment.

### 1.3. Research Gap and Key Contributions

To optimize existing MTD and Decoy deception management strategies by organization and overcome their limitation of low practical scalability, the objective of this study was to present PBNE (perfect Bayesian Nash equilibrium) and BSSG (Bayesian stochastic Stackelberg game)-based zero sum game foregrounds considering characteristics of real organizations and existing SOD2Gs and systematic deception game frameworks based on the POMDP (partially observable Markov decision process) state-transition background by attack model and defense model. This study will also simulate concepts of limited asymmetric rewards, views, and share manipulation according to defender's disinformation-based false information push. Adaptive deception decision-making was also characterized to decoy attackers' cognitive bias according to the defender's goal. This study also calculated a general-purpose-defined organizational scenario based on the proposed architecture, conducted simulations based on the CKC sequence as well as deception metrics and parameters for comparison, and calculated the optimal value of deception according to evaluation and verification results.

Major contributions of this study are as follows:

- Game theory was used to formulate an MTD that newly defined defender dominant deceiving behaviors according to the selection of a mutation cycle by accommodating attacks with loose adaptive mutations instead of strong prior avoidances as well as OSINT-based decoys by organization that could maintain decoying, isolation, and continuous deception resulting from the attackers' initial cognitive bias.

- By applying social engineering characteristics by organization to customized MTD (LPC-MTD) and Decoy (HS-Decoy), measures to strengthen prior preventiveness and post responsiveness within the organization were conceptually secured and deception efficiency was calculated based on the open organizational OSINT.

- PBNE and BSSG-based zero-sum game models universally composed by scenario were used to simulate real-time attack-defense competition within an organization with limited resources and model a multi-staged spatiotemporal deception decision-making process. In addition, by combining POMDP state-transition models, limited views by attacker and defender along with related attack surface concepts were simulated. The defender can also push disinformation-based deception signals for any single host initially intruded upon or finally occupied to the attacker as well as inverse transfer false host occupation information to the attacker to make the attack chain biased to be defender dominant or make it stay in the sandbox for a long period of time.

- When responding to in-depth attackers by organization, it is possible to establish an optimal defensive deception that minimizes operational performance degradation and maximizes security while operating independently without separately applying additional procedures or dedicated protocols based on scenarios and metrics.

The composition of this paper is as follows. First, Section 2 presents preceding game theory studies on MTD and Decoy concepts of defensive deception and compares them with the newly proposed concept of organizational deception games. In Section 3, with the goal to address limitations of existing MTD and Decoy, to evaluate and optimize real-time decision-making strategies between attackers and defenders by actually defined organizational environment, PBNE- and BSSG-based zero-sum game foregrounds and SOD2Gs (which are POMDP state-transition background-based organizational deception game frameworks) in practical application are described with an internal attack-defense

model. Related game metrics, formulas, social engineering knowledge, and deceptive parameters are then detailed. In Section 4, multiple organizational scenarios in SOD2Gs are specified together with CVEs and CVSSs by host to configure topology test beds, a social engineering attack-defense competition is simulated, and topology test beds are compared and analyzed along with an analysis of sensitivity by metric. In Section 5, inherent limitations of this study are discussed in addition to improvement measures. Finally, conclusions are drawn in Section 6.

## 2. Related Studies and Taxonomy of Proposed Deceptive Game Framework

Although MTD partially belongs to the scope of defensive cyber deception, unlike Honey-X or Decoy, it neither projects false information to actively mislead the attacker nor conceptually involves any scheme that actively induces the attacker's cognitive bias according to the defender's intention. In other words, their show crucial differences in that MTD mainly efficiently changes and proactively avoids configuration of internal networks and system hosts to be protected while maintaining availability of major service functions for legitimate users, whereas Honey-X and Decoy manipulate attackers' perceptions while carrying out reactive induction and isolation [1].

Based on these differences and goals of the present study, designs in previous studies are categorized as 'Game-theoretic Deception with MTD' or 'Game-theoretic Deception with Honey-X and Decoy' to upgrade the proposed game framework and attack-defense model into concepts that can be applied in practice to improve their performances and optimize their equilibrium state.

### 2.1. Game-Theoretic Defensive Deception with MTD

The core of game theory studies using MTD involves competitively modeling the attacker's CKC tactics and the defender's proactive evasion tactics as well as optimizing MTD variables to achieve goals held by each player. The main purposes of such studies are to minimize performance degradation and maximize security (thereby maximizing the defenders' gains) and to advance deception strategies in order to minimize attackers' gains such as through lateral movements and occupation of the target. Such studies are divided into 'general game theoretic-based studies', 'Bayesian Stackelberg game theoretic-based studies', and 'stochastic game theoretic-based studies' [1,6,7].

First, among general game theoretic-based studies, Zhu et al. [38] have quantified the trade-off based on the defender's security enhanced by MTD and degraded operational performance by applying the general game theory-based sequential attack-defense competition formula and related metrics to the MTD mutation concept. This allows for the concept of MTD games related to attacker reconnaissance evasion and intelligence collection efficiency minimization to be established for the first time. Ge et al. [39] have proposed incentive compatible MTD games based on communication mapping between normal users during server migration as a methodology that could be used to ensure high network service visibility and throughput for legitimate users as well as improve cyber agility [40] in terms of availability. This methodology could compose the concept of MTD operation for securing service availability and cyber agility in an architecture defined in detail for the first time. Carter et al. [41,42] have proposed a dedicated game to secure a migration optimization strategy to maximize seamless connection to legitimate users' services while minimizing suspicion among attackers that have been induced and isolated into the sandbox. Results of their study have verified that platform diversity based on overlapping cloning for the application of MTD at the lower stage is effective for some exploit attacks and more efficient than pure random mutations. However, proposed methodology is inefficient for rapid local attacks and also has proven that ensuring high hierarchical diversity by dummy platform can yield higher proactive benefits than having a large number of dummy platforms horizontally.

Second, among Bayesian Stackelberg game theoretic-based studies, Hasan [43] has proposed co-resident attack mitigation and prevention (CAMP), which is a Nash

equilibrium game model for detecting co-resident attacks in a virtual environment where the same spatiotemporal resources are shared while minimizing the impact of internal and external threats in a compromised joint virtual environment. The proposed CAMP has been proven to be able to immediately provide active attacker induction and active isolation tactics related to the optimal MTD defense strategy in the VM made by cloning the normal server. Feng et al. [44] have proposed a Bayesian Stackelberg-based artificial information disclosure model. Based on the Stackelberg game for applying the leader and follower scheme and the signaling game, which is an interactive decision strategy, they have proven that intentional disclosure of false information by the defender is a rather promising methodology that can disturb and bias the initial decision of the attacker to attenuate the continuity of the attack while simultaneously enhancing the agility of the defender. In a follow-up study, Zhu et al. [45] have proposed a Stackelberg game framework that generates and publishes false packets characterized in the reconnaissance stage of internal and external attackers. The efficiency of the indeterminate attacker induction and isolation mechanism in the multi-routing network topology was improved in their study. Among related studies, their study secured the directionality of application of the deception concept according to the construction of a scenario model close to the practical working environment for the first time. Sengupta et al. [35,46] have proposed a Bayesian Stackelberg game to optimize the deployment of security systems in the web and cloud. With that game, it is possible to formulate an MTD strategy that maximizes the proactive security using the system configuration set candidate while minimizing the mutation cost and performance degradation rate of the defender with limited resources. In a follow-up study, Li et al. [47] have proposed a Markov Stackelberg game to calculate the spatiotemporal MTD mutation decision-making of the defender against advanced attackers and composed an optimization formula based on the average-cost semi-Markov decision process (SMDP) and the discrete time Markov decision process (DTMDP). With their game and formula, it is possible to empirically evaluate and verify the improvement of the MTD decision-making performance of the defender who proactively responds to the short-sighted adaptive professional attacker with better performance than previous related studies.

Third, among stochastic game theoretic-based studies, Manadhata et al. [48] have diversified the dynamics between attacks and defenses based on the concept of probabilistic transition according to the decision-making flow and proposed a game model that reflects the foregoing in the strategies by MTD mutation state as an extension of previous studies regarding the diversification of attack surfaces and exploration surfaces. Their study showed that it was possible to formalize the MTD equilibrium concept and related trade-offs for modeling the optimal pro-active defense strategies based on dynamic surface compositions by scenario environment.

### 2.2. Game-Theoretic Defensive Deception with Non-MTD

The core of game theory studies using Honey-X and Decoy involves modeling attackers' detour tactics and defender's reactive defense tactics and conceptualizing decoying and sandboxing to realize disturbance, induction, and isolation based on signals corresponding to certain requests. The main goal of such studies is to present a strategy that can minimize performance degradation and maximize the efficiency of deception. In this subsection, 'game theoretic with MDP and POMDP studies', 'signaling game theoretic studies', and 'static game theoretic studies' are described briefly.

First, among studies on game theory with MDP and POMDP, Bilinski et al. [49] have described solutions based on masking games and Stackelberg games that can hide the essence of the defender node through processes such as inquiries between attackers and defenders, payment of costs, and selection of the final compromise target. These solutions make it possible to probabilistically analyze the attacker's potential behaviors before and after the exploit. They can also make adaptive configurations of the defender's deception strategies by episode. Durkota et al. [50] have proposed an MDP and sibling-class

pruning-based Stackelberg game to optimize deception strategies in the form of searches by time point of the time to the defender who deploys a honeypot to induce an attacker who then complies with the CKC attack graph. Anwar et al. [51] have quantified a static game that combines POMDP and a stochastic game as a honeypot and decoy allocation methodology considering probabilistic uncertainty through attack-defense interaction and model attack graph-based decision making. In that game, related to ramifications of the honeypot and decoy that perform attacker cognitive bias and induction, the researchers constructed a deceptive container management plan that yielded the highest defendability and the lowest cost for defenders with limited utilities. They also presented a heuristic optimization for alleviating the complexity of deceptive container allocation according to the potential decision-making interaction between attack and defense.

Second, among signaling game theory studies, Pawlick [52] has investigated and analyzed multiple signaling game and modeled them to devise a related solution while considering conditions and situations in which an attacker detects and recognizes the defender's honeypot. That study proved that the attacker detection ability formulated based on the present situation of the defender's deception identified in advance or evidence collected according to the interacted signals by episode did not necessarily diminish the deception efficiency and security of the defender. It also verified that the concept of artificial exposure could be expanded and applied to other non-signal game-based deception models. In a follow-up study [53], the concept of artificial exposure has been improved as a signaling game framework that could probabilistically induce and make judgments to estimate evidence based on the sender type and transmitted message, thereby proving again that the concept is vulnerable to attackers who can analyze the signals. Mohammadi et al. [54] have proposed a signaling game for identifying and deceiving external attackers using false defender avatars to optimize decoy strategies and alert thresholds. In that study, they defined the uncertainty of the judgment of the identity of the defender by the attacker who received signals from a real defender or a false defender avatar and considered the concept of insider authorization based on access control in the system architecture. Casey et al. [55] have proposed a basic compliance signaling game with which to consider the interaction between an organization's defender and a hostile insider, who can acquire the surface of the organization at the beginning and attempt to infringe on all hosts. The researchers have established the concept of a honey surface that perturbs and isolates the cognitive bias of malicious insiders. A follow-up study [56] has proposed an advanced compliance signaling game to simulate the interaction between an attacker that adaptively attempts to compromise and a defender by learning the defender intelligence that changes as the insider threat within the organization deepens.

Third, among static game theoretic studies, Nan et al. [57,58] have proposed a sandbox-based mixed-type perfect Nash equilibrium game framework in which the defender intelligently selects the most likely false node to formulate a deception strategy that biases the attacker to attack the deceptive node while wasting resources. Using that framework, they presented the most efficient deception node deployment strategy to the defender with limited resources, as well as configure a reactive management plan to achieve isolation, access blocking, and exit the induced attacker along with a primary deception action. Dimitriadis et al. [59] have proposed 3GHNET as a non-cooperative zero-sum architecture that can prevent DDoS and node breaches and enhance the security of 3G core networks within an organization. Their study showed that it was possible to formulate a payoff matrix to assists the defender in constructing an optimal deception strategy through two gateways that could control and capture the data flow between the sending and receiving nodes as well as a Nash equilibrium calculation focusing on the network attacker. Huang et al. [60] have proposed multi-frameworks reflecting the static Bayesian game and the asymmetric information one-shot game to consider the covertness and social engineering deceitfulness of professional attackers. The proposed multi-frameworks were verified within the Tennessee Eastman-based APT scenario.

### 2.3. Analysis by Previous Studies for Proposed Deceptive Game Concepts

To alleviate and solve the above-mentioned limitations, SOD2G, a systemic deception game framework based on PBNE and BSSG-based zero-sum game foreground and POMDP state-transition background, together with internal attacker and defender models, and the analysis of their classification are shown in Table 1.

**Table 1.** Taxonomy of existing defensive deception research studies and proposed deceptive game framework.

| Approach | Specific Technique | Advantages | Disadvantages |
|---|---|---|---|
| Game-theoretic defensive deception with MTD [38–48] | General game, Bayesian Stackelberg game, Stochastic game | Formalizing the interactions between attackers and defenders and providing methodologies for estimating deception effectiveness and security. Determining the optimal MTD strategy based on learning and realizing the decision model. | Not working the best strategy devised by a defender for the irrational attacker. Very high computational resources used to derive the optimal solution. Causing uncertainty due to different interpretations of the same game depending on the subjective perception of each different player. Difficulty in modeling cybersecurity and defensive cyber deception issues in real-world and practical environments. |
| Game-theoretic defensive deception with non MTD [49–60] | Game with MDP and POMDP, Signaling game, Static game | Realizing improvements in intrusion detection and prevention schemes by collecting additional attack intelligence while protecting existing network and system components. Possesses high flexibility, applicability, and rationality for the application of hierarchical deception concepts. | Difficulty in developing realistic honeypots that can effectively deceive attackers in terms of complexity and cost with the advent of increasingly intelligent and sophisticated attackers. Difficulty in calculating optimal solutions for combining, managing, and deploying multiple elements of defensive cyber deception. Not considering the negative effect on legitimate users due to insufficient access control and authorization. Difficulty in modeling cybersecurity and defensive cyber deception issues in real-world and practical environments. |
| SOD2G framework | - Deception MTD with LPC-MTD, Decoy with HS-Decoy<br><br>- Social-engineering Open-source intelligence (OSINT), Disinformation<br><br>- Game-based Perfect Bayesian Nash equilibrium with zero-sum (PBNE), Bayesian stochastic Stackelberg game (BSSG), Partially signaling scheme for disinformation (PSG), Partially observable Markov decision process (PDMDP) | **Description and Improvement** | |
| | | LPC-MTD is an MTD strategy selection concept that deliberately tolerates high-level organizational attacks while avoiding automated attacks that cause noise. Using this concept, flexibility and scalability issues and systematic deception management strategy issues can be alleviated, and MTD's post-responsiveness can be secured.<br><br>HS-Decoy is an attacker cognitive bias induction concept composed of hierarchical OSINT and HUMINT-based false organization information. It uses the actual characteristics of each organization to increase the likelihood of an attacker's earlier attempt compared to other decoys, while minimizing the attacker's suspicion. Using this concept, it is possible to improve the post-responsiveness and scalability of decoys while alleviating the existing limitations of practical application.<br><br>The SOD2G framework is a two-player competitive game—which is composed of a PBNE and BSSG-based zero-sum game foreground and a POMDP state-transition background—which models the real time attack-defense process in the organizational environment with limited resources as a multi-stage temporospatial deception decision making process. With this framework, a limited view and surface can be constructed along with the disinformation deceptive signal push concept. | |

## 3. Proposed Organizational Deceptive Game Framework

The PBNE and BSS-based zero-sum game foreground and POMDP state-transition background-based systematic defensive deception game framework in this study is a measure that can alleviate all operational limitations, unrealistic decision-making issues, practical scalability issues, and so on in an organization environment of MTD and Decoy

concepts as well as normalize the deception optimality in each zero-sum-based competition scenario. Therefore, this section details the novel formulation by module and by the process of the characterized MTD and decoy concepts based on OSINTs by organization in addition to the attack-defense player model in the proposed SOD2G framework and related components. PBNE, BSSG, and POMDP-related deception-game metrics and equations are also defined with related tuples.

### 3.1. Concept of Social-Engineering Organizational Defensive Deception Game Framework

The major architecture of the proposed SOD2G is illustrated in Figure 1. First, the OSINT elements by organization collected and formulated with node-link-based information graphs in advance are applied to LPC-MTD and HS-Decoy (which are MTD- and Decoy-applied concepts, respectively) by major property and by function, leading to advancements in social engineering deception knowledge. Social engineering deception knowledge on which unique characteristics by organization are reflected by hierarchical element is then used as an interface between the game-based foreground and the state-transition-based background. First, in the game-based foreground, it is used as a preprocessed parameter for generating and updating deception strategies of the defender model in the PBNE- and BSSG-based zero-sum game components and signaling-based distribution behavior, as a variable to be referenced when plotting by metric in the optimizer module, and as a major template concept for the formation of the defender's countermeasure tactics in the attack and defense graph component. Further, in the state-transition-based background, it is applied as a deceptive signaling and decoying concept for the defender's microscopic view and occupancy rate, surface, and partial reward manipulation concepts in the POMDP. It is also used as an adaptive probability change factor in the transition probability matrix based on the vulnerability table in the transition component. Finally, it fosters all related concept of deception, competition between players, payoff tactics, and game equilibrium.

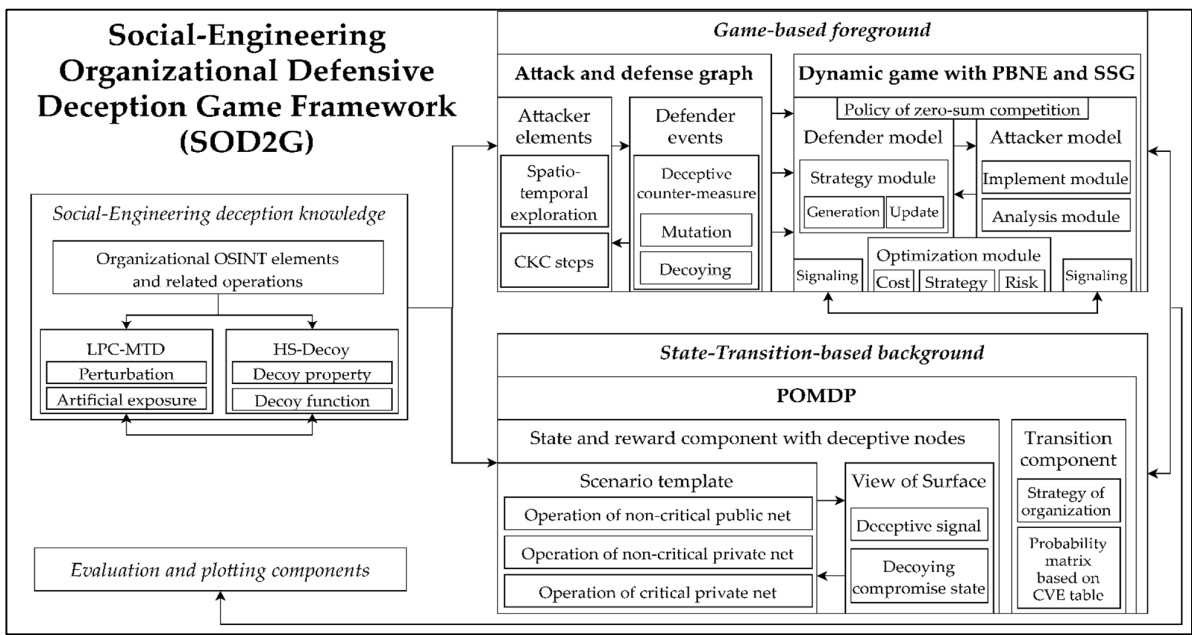

**Figure 1.** Main overview of the proposed organizational deceptive game with social-engineering MTD and Decoy.

### 3.2. Organizational Deception Knowledge with OSINT

Open-source intelligence (OSINT) is reliable information from open sources, government agencies, and so on. It goes beyond detailed information and characteristics to generally refer to even the collection process from open sources. It is mainly used in

various cyber and information security fields such as military, security, cyber security, infringement responses, disinformation, and deception-based intelligence. It is mainly classified as a domain-based category consisting of information flow on the Internet, public datasets maintained by governments, and private open datasets. Organizational OSINT [61] is characterized as the use of an information group consisting of unique OSINTs and rough fingerprints of foreign government military organizations collected and composed based on open sub-party services. It involves performing all necessary stages to proactively establish an OSINT strategy based on security requirements related to an organization's legacy operating environment mainly by identifying information sources, including a stage in which open OSINTs are collected based on identified organizational information sources and strategies, a stage in which acquired data are normalized based on correlation graphs and then supplemented based on parent-child nodes followed by a reinforcing step based on parent-child nodes, and a stage in which the reinforced OSINT information group is refined into social engineering deceptive knowledge and concretized as cyber space intelligence (CYBINT) in sequence. The composition of the related organizational OSINT information group is reduced and recomposed based on the network, host, and service layers as presented in Table A1 based on the concept of simulation of the view and surface of the attack-defense model in SOD2G.

*3.3. LPC-MTD and HS-Decoy for Organizational Defensive Deception Strategy*

First, loosely proactive control-based MTD (LPC-MTD) [61] is an organizational MTD concept designed to alleviate both conceptual limitations and social engineering scalability problems of the existing MTD. It artificially adjusts the mutation strength based on the intention of the defender to intentionally expose false information or present disinformation whereby the defender dominantly forces the attacker to make a hasty judgment as if he/she has successfully bypassed the defender's MTD interface. It also induces early cognitive biases of attackers, such as leading attackers to the defender's isolated sandbox by perturbing the defender's intelligence gathered by the attacker that has performed reconnaissance. The concept of LPC-MTD as such is defined with a focus on the Bellman value iteration-based batch sampling optimizer to select entropy-based adaptive MTD mutation strength according to changes in attack behaviors along with the perturbation [7] to asymmetrically impart noises to the attacker's cognitive directivity based on a limited view and related attack surface information. Based on that concept, along with additional advancement of the defender's deception strategies using the MTD cycle selection schemes by organization, MTD-based deception operation can also be formulated for network separation and closed network.

Due to the mobility-based MTD principle [6,11], the LPC-MTD is constructed as shown in Equations (1) and (2) for perturbation (*P*) and value iteration sampling (*VS*), respectively:

$$P = Pr[L = l | OS = \mu(os)], \tag{1}$$

where *L* is a set of virtualized IP, port, service, and host fingerprint elements that are similar to the actual information of the target to be protected in an organization, but are hierarchically composed so that only the defender and legitimate user can identify them. which are dynamically signaled under the leadership of the defender in order to bias the normalized gradient of the defender's intelligence possessed by the attacker toward the defender's advantage. OS is a set of OSINT information groups by organization. It is intended to improve the reliability of the virtual information generated for deception while minimizing the attacker's suspicion of disinformation and artificial information exposure through *L*. In Equation (1), $\mu(os)$) calculates the attacker's predictability for the secret being protected by the deceptive defender based on both observable arbitrary OSINT-based exposure factors *os* and the defender intelligence possessed by the attacker at present.

$$VS^n(i) = \min_{\tau_i, m_i \in M}\left[c_{i,j} + \sum \widetilde{m}_{i,j} VS^{n-1}(j)\right], \tag{2}$$

where $c_{i,j}$ is the mutation cost to shift or shuffle the surface index $j$ of the defender of the next episode with private information through an adaptive deception strategy of signaling a limited view and a false occupancy rate at the surface index $i$ of the defender of the current episode in order to optimize the trade-off of the legacy topology operating environment in an organization that has limited resources. $\widetilde{m}_{i,j}$ is the possibility to distinguish $i$ and $j$ following changes in $i$ and $j$. It means the possibility of minimizing the suspicion of the APT attacker who continuously observes the defender's environment. $\tau_i$ is the temporospatial resource consumed in relation to the intervention of the defender until the completion of the mutation of the mutation time slot length of the surface information related to $i$, and $m_i$ is the surface information sampled and optimized based on $i$ in the deceptive surface set $M$ of the defender, which can be created through mutation.

HS-Decoy (hierarchical social engineering decoy) [61] is also an OSINT-based organizational decoy concept that either statically induces attackers to continue to attack a false target that is not a target of protection to further improve the deception efficiency within an organization of the existing decoy or forces successive attack chains of attackers that have been isolated to remain in the sandbox while deceiving the attackers to stop them from attempting to escape. Since it mainly appears similar to normal service or important information while being more enticing based on OSINTs by organization, it looks valuable to attackers. However, it has a complex multitenant-based architecture composed of a large number of false data in reality, thus enabling legitimate users within the organization to distinguish the foregoing from real information while operating such that attackers cannot distinguish it from real information without prior knowledge of the target system. It also increases hierarchies and number of entities to statically deceive attackers using layered OSINT elements while independently improving redundancy and diversity concepts based on detailed deception elements in the decoy to minimize suspicion of the attacker who invades in the early stage in relation to the parallel defender environment recognition and verification behavior of the attacker. The concept of HS-Decoy is therefore upgraded by combining hierarchical OSINT information groups by organization together with detailed complex properties such as 'reliability and distinguishability', 'attack inducibility', 'permanent exposability', 'diversity', 'redundancy and non-coherence', 'distinguishability', 'detectability', 'role-based access control', and so on to induce continuous attacker inferiority in relation to access event count, intrusion tolerance and restoration, induction of static attackers' cognitive bias in the early stage, isolation, and so on. Through the foregoing, for dynamic attackers' cognitive bias in the early stage and intentional permitting of invasion, which is primarily performed at the network and host level, HS-Decoy, which has been secondarily combined, can continuously force defender predominance in the isolation environment using independent distributed multi-tenant based multiplexed OSINT-based decoys with cloning and mimicking.

As such, the HS-Decoy will focus on the property-based decoy principle [9] based on Equations (3)–(9):

$$Pr\left[Exp_{A,H,O}^{believe} = 1\right] \leq \tfrac{1}{2}, \tag{3}$$

First, Equation (3) is Believability (B). It calculates the attacker $A$'s trustworthiness of HS-Decoy, who does not have information on the internal distinction criteria related to the protection target. and for the hierarchically formulated HS-Decoy set $H$ based on both $O$, which is a set of organizational OSINT elements reconstructed for social engineering decoy operation, and information on the unique characteristics of the actual protection target. To elaborate, when A composes an arbitrary defender's surface

information, it is judged to be an active defender deception, and cases where $H$, which is defender dominant, is excluded as a stochastic noise are guaranteed to be 0.5 or less.

$$Pr[o \rightarrow O | o \in PF] = Pr[h \rightarrow O | h \in H], \qquad (4)$$

Equation (4) is Enticingness (E). It normalizes the attacker's hasty judgment such as IP, port, service, host fingerprint, and so on from $o$, which is an arbitrary element of organizational OSINT for decoy operation in $O$, to $PF$, which is an index for the degree to which attackers are quickly forced by the defender dominantly to calculate the possibility of static signaling to intentionally induce an infringement action on the $o$-based $h$. It also verifies that characteristics of $PF$ in $h$, which are created, distributed, and managed by the deceptive defender, do not deviate from the uniqueness of $o$, which minimizes the attacker's suspicion based on gradual mutation.

$$\prod_{i=0}^{n} Pr[V_i] > \delta, \qquad (5)$$

Equation (5) is Conspicuousness (C). It calculates the possibility of static dazzling for the limited view $V_i$ related to the $i$-based surface information of the attacker and defender formulated in the current episode based on mutual signaling through a comparison with $\delta$, which is an index standardized for the degree of salience of factors causing extreme bias in the formation of continuous CKC chains such as CVE vulnerability among $o$. That is, beyond passively advertising only CVE vulnerability information to the attacker, it also guarantees the execution of $V_i$-based projection to additionally prove the false reliability of the relevant vulnerability information to the attacker.

$$Pr[h \rightarrow O: CD_{A,h} = 1] \geq \epsilon, \qquad (6)$$

Equation (6) is Detectability (DE). It calculates the detectability for $CD_{A,h}$, which is an index for the degree to which the act of attacker $A$ infringes on $h$ through comparison with the detection threshold $\epsilon$ in $h$ for adaptive management and for updating of $h$ according to changes in the external entropy. It also ensures the possibility of false positive and false negative to improve the defender's deceivability according to the decoy beacon and emulator.

$$Pr[h \rightarrow H: Exp_{A,H,O,h}^{believe} = 1] \leq \frac{1}{2}, \qquad (7)$$

Equation (7), which is based on (3), is Variability (V). It calculates the trustworthiness of $h$, a single HS-Decoy element, in a certain $H$. That is, by inducing the attacker's cognitive bias and securing different distinguishability for all $h$ in $H$ where the decision gradient will be defender dominantly perturbed, it ensures the enhancement of hierarchical deception diversity for each $h$.

$$Pr[CT_{D,o,h} = 1] = Pr[CT_{D,o,h} = 1 | H], \qquad (8)$$

Equation (8) is Non-Interference (NI). It calculates the possibility of reactive non-interference for the contiguity of defender $D$ (a legitimate user in the organization for $o$-based $h$) to be clearly distinguished as a sender for deceptive signaling while immediately being treated as an exception so that it would not be selected as a decoying target. That is, it guarantees role-based access control schemes that are divided such that attackers can access only $h$ where legitimate users and $D$ can only access the real protection target.

$$Pr[Exp_{D,H,O}^{believe} = 1] = 1, \qquad (9)$$

Finally, Equation (9) is Differentiability (DI), which is based on Equations (3), (7) and (8). It calculates both the distinguishability of defender $D$ and the indistinguishability of attacker A for $O$-based $H$ and the real protection target. That is, the protection target cannot be distinguished from $H$ with a secondary deception policy if the attacker

understands the abstracted organizational protection target in the isolation environment, which the attacker cannot reconnoiter or explore, through primary deceptive signaling such as MTD.

Finally, utilizing all organizational OSINT information groups, LPC-MTD, and HS-Decoy, they are normalized with the social engineering deceptive knowledge in SOD2G. They are then applied as major atomic input parameters in each of PBNE- and BSSG-based foreground and POMDP-based background components.

*3.4. Construction of Organizational Dynamic Deceptive Game Framework*

As shown in Figure 1, the game-based foreground in SOD2G consists of a zero-sum-based dynamic game component based on PBNE and BSSG, clustered elements for performing the next action based on transition probabilities by player of the transition branches by state in the POMDP, and attack and defense graph components, which represent an event information group.

3.4.1. Design Principles with Deceptive Multi-Player Game Architecture

Attack and defense graph components are configured as a threat model for the attack flow to achieve occupation of the final intrusion or suboptimal intrusion target point based on the temporospatial search and CKC stages formulated by the organizational scenario predefined in the scenario template component with POMDP-based background. The dynamic game components adopt the PBNE game strategy to optimize the judgment that can maximize the payoffs by episode for private asymmetry based on incomplete information by player. They also adopt BSSG strategies based on the leader and responsive followers. Ripple effects of signaling by player are also added to apply a quantitative sequential relationship for partial signaling between the attacker and the defender stochastically based on the concept of compensation.

In this case, dynamic game components, which are configurable based on Figures 1 and 2, are composed in detail while centering on the following 9-tuples.

1.  $N = (N_A, N_D)$ is a set of players where $N_A$ is an attacker and $N_D$ is a defender. In this case, depending on the direction of the payoff, signaling, and feedback by player in a random episode, $N$ is formulated with the attacker as an active APT leader and sender/defender as a passive reactive follower/receiver, or conversely, with the defender as a dynamic deceptive leader and sender/attacker as a passive naïve follower/receiver.

2.  $TS = (TS_{N_A}, TS_{N_D})$, $TS_{N_D} = (ts_i | i = 1,2, \dots, n)$, and $TS_{N_A} = (\rho)$ are sets of player types. $TS_{N_D}$ is defined as an element of the hierarchical deceptive and defense graph-based private information of defender $N_D$. $TS_{N_A}$ is defined as an element of the attack-graph-based private information of attacker $N_A$. Player types are divided or combined with abstraction based on abilities to subtract and add rewards by player. And the attacker to the defender who has non-deterministic elements decisively composes elements with the defender intelligence validity, which is termed $\rho$.

3.  $GS = (GS_{N_A}, GS_{N_D})$, $GS_D = \left( gs_{N_{d_i}} \middle| i = 1,2, \dots \right)$, and $GS_A = \left( gs_{N_{a_j}} \middle| j = 1,2, \dots \right)$ are sets of game strategies related to mutual zero-sum competition between individual attackers $N_A$ and defenders $N_D$. They are composed according to the sender and receiver signaling relationship. $GS_D$ is the deceptive and defense graph-based strategy possessed by the defender. $GS_A$ is defined as the attack-graph and intelligence-based strategy possessed by the attacker as valid defender surface information.

4.  $SS = (SS_{N_A}, SS_{N_D})$, $SS_{N_D} = \left( ss_{N_{d_i}} \middle| i = 1,2, \dots \right)$, and $SS_{N_A} = \left( ss_{N_{a_j}} \middle| j = 1,2, \dots \right)$ are sets of signals of attacker $N_A$ and the defender $N_D$. They are selected or released according to signaling mechanisms, which vary between active and passive from

player to player. The attacker has $SS_{N_A}$ as a leading attack signal set to achieve the final invasion target. The defender has $SS_{N_D}$ as the leading deceptive and defense signal set for complete blockage and exit of the attacker.

5.  $S = (s_i|i = 0,1, \dots k)$ is a finite set of states based on GS and SS in game components. It defines multi-level and transitivity in the attack-defense competitive game environment along with actions.

6.  $A = (A_{N_A}, A_{N_D})$, $A_{N_D} = \left(a_{N_{d_i}}^i \middle| i = 1,2, \dots x\right)$, and $A_{N_A} = \left(a_{N_{a_i}}^j \middle| j = 1,2, \dots y\right)$ are finite sets of actions of attacker $N_A$ and defender $N_D$ for $S$. $A_{N_D}$ defines the defender's deceiving, defending, or false negative actions for $s_i$ as a transitive relationship. $A_{N_A}$ defines attacker's actions for $s_i$, such as reconnaissance, search, vulnerability and fingerprint-based exploits, early occupation, lateral movement, and final invasion through target point privilege elevation and takeover.

7.  $\theta\left(S_k, a_x, a_y, S_{k^`}\right)$ is a probability distribution function used to calculate the probability of reaching $S_{k^`}$ in the case where attacker $N_A$ performs the action termed $a_x$ and defender $N_D$ performs the action termed $a_y$ in the current episode $S_k$.

8.  $R(S_k, a_x, a_y)$ is a function used to calculate the reward that can be obtained from the player's judgment in the episode when attacker $N_A$ and defender $N_D$ perform actions termed $a_x$ and $a_y$, respectively, in $S_k$. In response, players compete in the direction to maximize $R$'s reward.

9.  $U = (U_A, U_D)$ is a signaling-based discount factor function. It cuts off the judgment ranges by player within [0,1] to attenuate effects of signaling actions by player. It also simulates pre- and post-competitive strategy judgments by player in the leader-follower relationship, such as limited views and the concept of true-false shares.

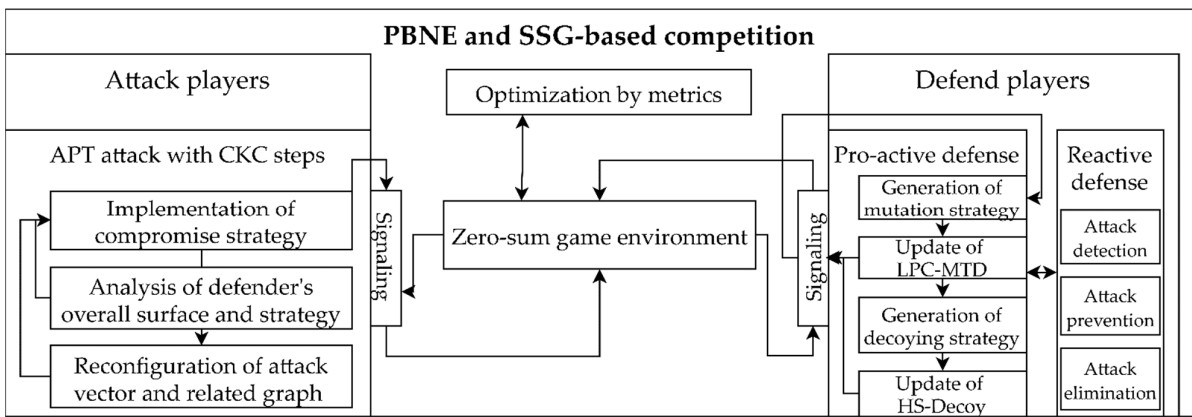

**Figure 2.** Detailed overview of zero-sum-based attack-defense competition with PBNE, BSSG, and signaling.

As specified in the nine-tuples above, the signaling performed by each player in SOD2G is defined based on the signal game strategy to perform multistage-layer-based information transmission and BSSG. Causality of transmission and reception is determined according to the optimization directionality of payoff by player and the signaling initiative. It is terminated as a PBNE-based optimized equilibrium state.

### 3.4.2. Configuration and Optimization of Zero-Sum-Based Attack-Defense Competition

First, the attacker, who is a sender and leader from the viewpoint of attack actions by episode, delivers signals related to concepts of early reconnaissance and search, exploitation, privilege escalation, and host occupation to the victim host in the defender model or the defender dominant deception environment. In response to signals, the passive defender, who is a receiver and follower, dynamically performs feedback according to the concept of a vulnerable point of contact or a pre-configured deceptive

sandbox that does not consider defense in advance. Next, the attacker verifies the validity and reliability of the defender intelligence and surface, view, and occupancy information possessed at the present time through the naive or deceptive response of the defender to the signals. The attacker then accordingly reconstructs its invasion strategies, attack vectors, and so on. In this case, the attacker's judgment range is cut-off spatiotemporally according to a predefined discount factor to induce reduction of solution space and derivation of approximate values to calculate the optimal value of the total reward related to the invasion target. In addition, based on the maximum number of intrusion attempts, which is a private information-based conceptual factor to achieve the attack goal, the payoff that is inversely added by the defender dominantly due to the achievement of attacker dominant asymmetry or the attacker' wrong judgment is calculated in terms of utility and cost or reward.

Next, the adaptive defender, who is an active deceptive sender and leader from the viewpoint of defensive actions by episode, forces defender dominant early cognitive bias based on disinformation operation to the attacker before or during the process of carrying out reconnaissance and search. As well as delivering deceptive signals to deceive the attacker so that the attacker is induced to a false environment as invasion strategy reconstruction input values in the attacker model. In response to signals, the passive attacker, who is the receiver and follower, dynamically performs the feedback to update the defender intelligence possessed by the defender dominantly without doubt. Through the response of the attacker who accepts the signals, the defender then evaluates the efficiency and scalability of the proactive MTD as well as the reactive decoy-based defensive deception strategy being followed at the present time. It additionally updates layered deception strategies and host defense elements according to results. At this time, the defender's judgment range is also a cut-off according to a predefined discount coefficient, thus yielding a payoff approximate value that is not completely optimized by episode.

The maximization of reward through reasoning between leading players in the leader-follower relationship based on signaling by episode is organized as a Q-value as shown in Equation (10). In this case, $U$ and $TS$ are defined as players with signaling initiative in the current episode:

$$Q\left(S_k, a_x, a_y\right) = R\left(S_k, a_x, a_y\right) + U \sum_{S_{k`}} \theta(S_k, a_x, a_y, S_{k`}) \cdot TS \cdot OPT(S_{k`}), \qquad (10)$$

That is, $OPT(S_{k`})$, from the viewpoint of the attacker who is actively signaling, is expressed as Equation (11) through $SS$, which is the signaling action that can be performed in $S_{k`}$. This produces the optimized maximum reward with judgement based on incomplete and private information:

$$OPT(S_{k`}) = \max_{SS} \min_{a_x} \sum_{ay} Q\left(S_k, a_x, a_y\right) \cdot \left(ss_{N_{d_i}}\middle| i = 1,2,\dots\right),, \qquad (11)$$

Next, the partial signal game-based PBNE in the game environment based on $OD$ and $OA$ is generally summarized as shown in Equations (12)–(15) based on Equations (10) and (11). In this case, $P_D$ in Equation (12), which is similarly constructed based on $OPT2$, is the prior probability-based defender's judgment probability for $SS_{N_A}$-related $TS_{N_D}$. $P_D'$ *in* Equation (13) is the defender's inference probability based on the posterior probability related to $SS_{N_D}$ reconstructed based on the updated internal deception-defense strategy after the feedback-based signaling of the defender for $SS_{N_A}$. In addition, when calculating PBNE through $P_D$ and $P_D'$ as such, it is affected by $U$, a discount factor. The PBNE state entry in game is also controlled according to the configuration of $U_A$ or $U_D$ based on whether or not the signaling leader is selected:

$$P_D = \left(p_D \cdot (TS_{N_{D_i}})\middle| i = 1,2,\dots n\right), \qquad (12)$$

$$P'_D = P'_D \left( \left( TS_{N_{D_i}} \middle| i = 1,2,\dots n \right) \middle| SS_{N_A} \right), \tag{13}$$

$$OD(SS_{N_{A_j}}) = arg \max_{SS_{N_{D_k}} \in SS_{N_D}} \sum_{TS_{N_{D_i}} \in TS_{N_D}} P'_D \cdot F(TS_{N_{D_i}}, SS_{N_{A_j}}, SS_{N_{D_k}}), \tag{14}$$

$$OA(TS_{N_{A_i}}) = arg \max_{SS_{N_{A_j}} \in SS_{N_A}} F(TS_{N_{A_i}}, SS_{N_{A_j}}, OD(SS_{N_{A_j}})), \tag{15}$$

Finally, to reflect the zero-sum based attack-defense competitive game presented in SOD2G above, it is structured as an interface as shown in Figure 3 based on the state, transition, and arrival concepts in POMDP components specified in Figure 1.

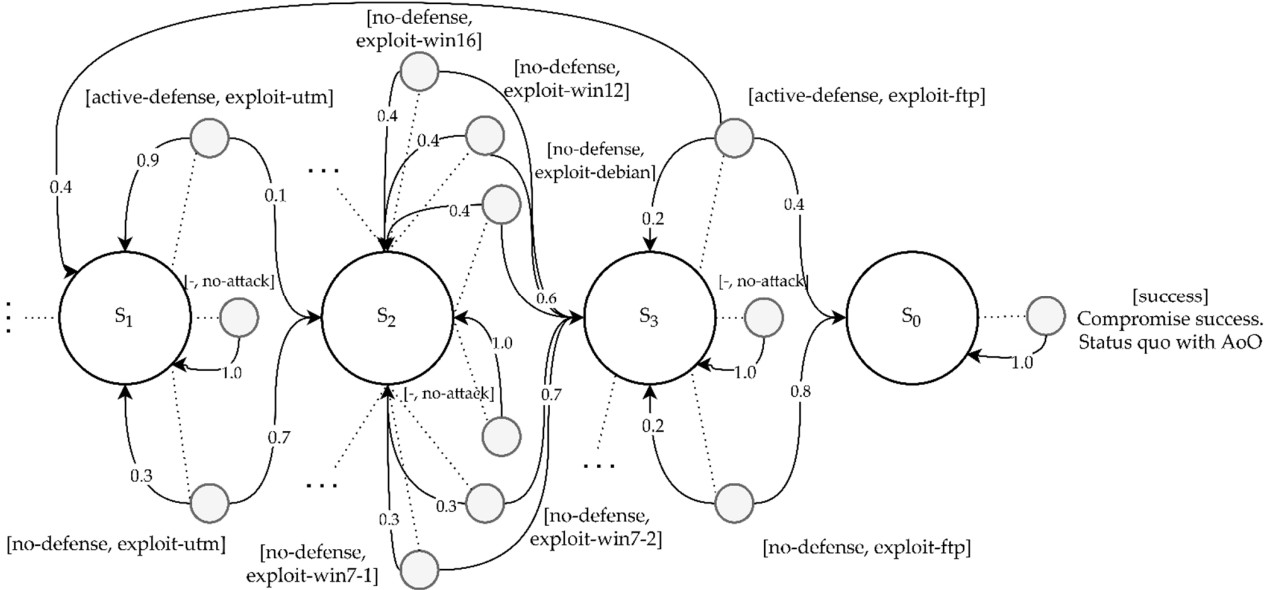

**Figure 3.** Sub-overview of POMDP with attacker side in Scenario 1.

Transition probability and reward setting in the POMDP are also detailed in Table 2, although they are configured differently according to pre-defined scenarios.

**Table 2.** Probability of transition and semi-constant reward value with each state and action in Scenario 1.

| State | Probability of Transition | | | | | | Reward Value for Defender | | | | | |
|---|---|---|---|---|---|---|---|---|---|---|---|---|
| $S_0$ | $[(1,0,0,0)]$ | | | | | | $[-20]$ | | | | | |
| $S_1$ | $\begin{bmatrix} (0,1,0,0) & (0,1,0,0) \\ (0,0.3,0.7,0) & (0,0.9,0.1,0) \end{bmatrix}$ | | | | | | $\begin{bmatrix} 0 & -3 \\ -10 & 10 \end{bmatrix}$ | | | | | |
| $S_2$ | $(0,0,1,0)$ | $(0,0,1,0)$ | $(0,0,1,0)$ | $(0,0,1,0)$ | $(0,0,1,0)$ | $(0,0,1,0)$ | $0$ | $-3$ | $-3$ | $-5$ | $-5$ | $-5$ |
| | $(0,0,0.3,0.7)$ | $(0.4,0,0.4,0.2)$ | $(0.8,0,0.1,0.1)$ | $(0,0,0.3,0.7)$ | $(0,0,0.3,0.7)$ | $(0,0,0.3,0.7)$ | $-9.3$ | $9.3$ | $-3$ | $-5$ | $-5$ | $-5$ |
| | $(0,0,0.3,0.7)$ | $(0.4,0,0.4,0.2)$ | $(0.8,0,0.1,0.1)$ | $(0,0,0.3,0.7)$ | $(0,0,0.3,0.7)$ | $(0,0,0.3,0.7)$ | $-10$ | $-3$ | $10$ | $-5$ | $-5$ | $-5$ |
| | $(0,0,0.4,0.6)$ | $(0.45,0.15,0.15,0.25)$ | $(0,0,0.4,0.6)$ | $(0,0,0.4,0.6)$ | $(0.8,0,0.1,0.1)$ | $(0,0,0.4,0.6)$ | $-7.5$ | $-3$ | $-3$ | $7.5$ | $-5$ | $-5$ |
| | $(0,0,0.4,0.6)$ | $(0.45,0.15,0.15,0.25)$ | $(0,0,0.4,0.6)$ | $(0.8,0,0.1,0.1)$ | $(0,0,0.4,0.6)$ | $(0,0,0.4,0.6)$ | $-10$ | $-3$ | $-3$ | $-5$ | $10$ | $-5$ |
| | $(0,0,0.4,0.6)$ | $(0.4,0.2,0.2,0.2)$ | $(0,0,0.4,0.6)$ | $(0,0,0.4,0.6)$ | $(0.8,0,0.1,0.1)$ | $(0,0,0.4,0.6)$ | $-10$ | $-3$ | $-3$ | $-5$ | $-5$ | $10$ |
| $S_3$ | $\begin{bmatrix} (0,0,0,1) & (0,0,0,1) \\ (0.8,0,0,0.2) & (0.4,0.4,0,0.2) \end{bmatrix}$ | | | | | | $\begin{bmatrix} 0 & -3 \\ -10 & 20 \end{bmatrix}$ | | | | | |

## 4. Experiments and Sensitivity Analysis

In this section, the attack-defense scenarios in the SOD2G framework are formulated according to the concept of organizational operation due to the normalization scheme of conflict of game caused by the independent Monte Carlo based on true random generation as well as the fluid intrusion probability distribution between the attacker and the defender. Along with a comparison of the efficiency of zero-sum-based attack and

defensive deception according to successful intrusion and detection-induction-isolation-blocking success by node in the configured scenario, related sensitivity analysis is also performed in a general manner to ultimately calculate the relevant deception optimal value.

*4.1. Definition of Experimental Bayesian Stochastic Game-Based Scenarios and Related Metrics*

To experimentally verify the efficiency of defensive deception, complex metrics, attack-defense scenarios, and topologies related to the host-switch-security solution-based organizational legacy network operation are calculated as shown in Figures 4–6. Each scenario is formulated in detail with operational equipment and topology structure diagrams, spatiotemporal performances and costs by participant, invasion-defense discrimination criteria, CKC, attacker's and defender's final goals, attack-defense and deception sequence, other assumptions such as CVE and CVSS, and so on. Variables such as surfaces of attackers and defenders with limited views, counter-measure strategies, limited resources between service availability and security, real-time share competition related to successful invasions and defenses by node, disturbances due to deceptive signals, and so on are also implicitly applied by scenario and subdivided as listed in Tables A1–A3.

Accordingly, the formulated organizational operation scenarios have the following performance premises in common. A universal concept of attack-defense competition interface is established based on these premises.

1. Attack Behavior Standard: The attacker performs an attack on the organization's legacy internal network from the outside and selects either single or multiple final invasion target points to proceed with the invasion according to the episode. The selection of target points by episode differs depending on the intelligence of the defender possessed. The formation of attack chains continues in the direction that yields the highest partial gain.

2. Attacker's execution of professional APT attacks: The attacker gives the top priority to the final occupation of the defined single or multiple target points and maximizes the attack gain while optimizing all possible attack-based branches without any fixed attack path. The attacker also has the flexibility to select another host as the next best attack target such that invasion origins can change by episode immediately before the failure. Lateral movement can also be activated along with a change in the direction of movement in cases where the attacker fails to achieve the first priority target. However, the attack efficiency may decrease due to the deceptive false surface information of the defender. It may be impossible to judge Decoy and Honey-X based sandbox based on biased view selection for a long time.

3. Defender behavior standard: The defender carries out monitoring, reactive response, blocking, and exit of all hosts and terminals. However, resources for the operation of the defense solution and the execution of the adaptive active deception sequence are limited. Possession of prior information on the invasion target points is also limited. If counter-measure response points are selected incorrectly, the defense will fail and the general defense gain will be reduced. The defense goal of the defender is established differentially according to importance levels and potential vulnerabilities by node in the topology. However, the defender operates with a focus on successfully defending the occupation of all nodes and finally blocking the attacker. According to a separate scenario, a host invaded by an attacker can be restored by a cloned backup copy and formulated as a node occupied by the defender. Backup copies cloned by the host are snapshots after performing certain episodes. The process of isolating and storing them in an environment that can be invaded by attackers is executed.

4. Defender's performance of defensive deception: The defender performs MTD and Decoy-based proactive deception in the topology based on signaling. However, since only a single host and one terminal can perform reactive response to the attacker that

bypasses the deception and successfully invaded, an appropriate prevention point and a response point deployment strategy are needed to maximize limited resources.

5.  Reward Standard: The attacker acquires rewards by episode based on the success of the CKC stage and the defender acquires rewards based on the attacker's CKC protection success-exit-punishment. Through the foregoing, the reward is optimized into the maximum total reward value at the end of the final simulation.

6.  Constantization and variabalization of rewards: The concept of rewards in the SOD2G framework is a constant. It is established based on CVE and CVSS. However, according to the scenario and topology structure selection, it is also converted into a variable that is different according to the attacker's APT level and the defender's deception-response level. The probability of transition between states is also set as a constant. However, the attack-defense level is fixed at a certain level. The constant can also change dynamically depending on the present situation of the state of attack-defense shares.

7.  Encouraging rapid decision making: The micro-decision gradient is maximized according to successive spatiotemporal changes through a discount coefficient to prevent the fixation of the macro-equilibrium state. That is, the attacker and defender cannot indefinitely postpone the current decision for more than tens of episodes. Branch calculation space for obtaining the maximum reward at the present time is also limited in scope. This is carried out with a cut-off MILP optimization scheme.

8.  Scenario ending condition: The attacker's ultimate goal is to perform a CKC-based action of object (AoO) by reaching the currently established single or multiple invasion target point after the final exploit. The defender's ultimate goal is defined as depleting the attacker's APT attack attempts and related attack resources to force the attacker to completely exit from the current organization's internal network, finally punishing the attacker. Upon completion of the simulation, rewards by episode and attack-defense probability values are returned.

9.  Addition of the concept of security solution: unified threat management (UTM) is an integrated secondary security device in which IDS and IPS are mixed. It is performed for the purpose of attacker threat detection, blocking, and expelling. Firewall is a form of primary security equipment for user access control and the judgement of whether a user is legitimate, which is performed for the purpose of user validation, authentication, and authorization. In this way, security solutions are also upgraded, with a focus on horizontal diversity or vertical redundancy. In this case, potential vulnerabilities to define the APT attacker to penetrate from an external network to the organization's internal network through relevant security solutions are assumed as CVE and CVSS. Unlike other hosts, the attacker's acquisition of administrator privileges in security solution is limited not to be carried out.

Based on the foregoing, attack-defense scenarios are established as follows. Independent focus points related to social engineering deception and organizational legacy operation strategies are also defined in detail.

-   Scenario 1: 'Organizational open network based on a single security solution in a non-critical domain'

    (1) Host configuration: Local mail host (Win7-1), remote mail host (Win7-2), mail server (Win server 16), DB server (Debian), and internal network DNS server (Win server 12).

    (2) Security equipment configuration: Integrated single UTM with some firewall functions.

    (3) Attacker's goal: The best goal is to achieve DB hijacking-destruction after PFTPd remote code-based privilege escalation and acquire DB administrator privileges. The second-best goal is to carry out differential intrusion behaviors by host according to the level of vulnerability when it is judged that Debian-FTP cannot be compromised.

(4) Defender's goal: After securing the protection of hosts and terminals in all topologies, depleting all resources possessed by the APT attacker to achieve complete exit from the internal network.

(5) Major attack sequence: Outside the topology → Attempt to break into the internal network → UTM infiltration and bypass → Intrusion and occupation of many other hosts are carried out concurrently when immediate access to Debian-FTP and search is not possible, → Intrusion of Debian-FTP and achievement of final occupation.

(6) Major defense sequence: Perform proactive deception and monitoring of the entire topology → Perform attacker intrusion detection, reactive blocking, and exit → Achieve complete exit to the outside.

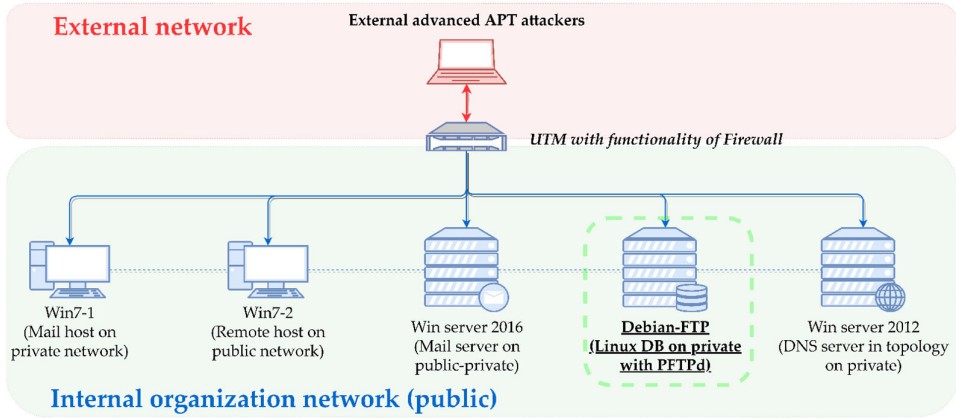

**Figure 4.** Overview and related topology with an organizational attack-defense strategy for Scenario 1.

- Scenario 2: 'Independent multi-security solution-based non-critical organizational semi-closed network'

(1) Host configuration, defender's goal, and major defense sequence: same as [Scenario 1].

(2) Security equipment configuration: horizontally diversified UTM and firewall.

(3) Attacker's goal: The best goal is to elevate privileges through heap buffer overflow remote code execution attack and obtain DNS server administrator privileges and achieve worm-centered network intra-control point occupation. The second-best goal is to conduct differential intrusion behaviors by host according to vulnerability when it is judged that malicious invasion through DNS service query in Win server 12 is impossible.

(4) Major attack sequence: Outside the topology → Attempts to break into the internal network → Infiltration and bypass according to UTM or firewall function → Intrude and occupy many other hosts concurrently when immediate access to Win server 12 and search is impossible, → Win Server 12 invasion and achievement of final occupation.

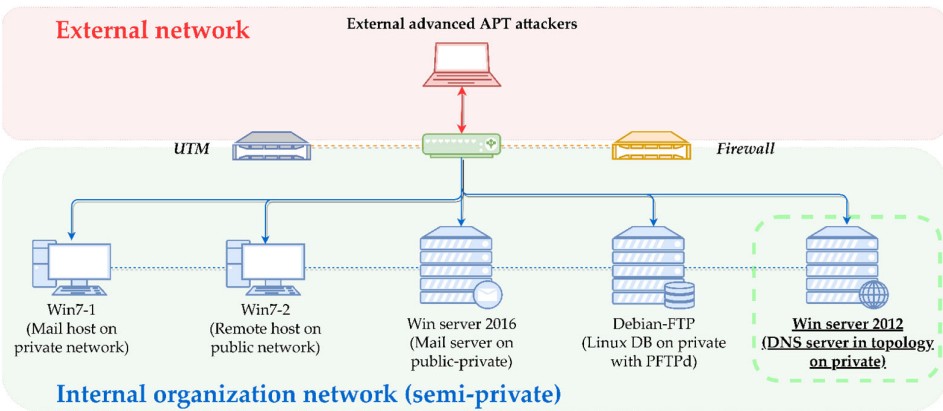

**Figure 5.** Overview and related topology with an organizational attack-defense strategy for Scenario 2.

- Scenario 3: 'Closed organizational network based on triple security solution in critical domain'
  (1) Host configuration: Additional definition of duplicated backup copies based on cloning of hosts in [Scenario 2]. This includes all live and unlive hosts.
  (2) Security equipment configuration: Three vertically redundant integrated UTMs and three firewalls.
  (3) Attacker's goal: The best goal is to obtain DNS server administrator authority through a heap buffer overflow remote code execution vulnerability attack, occupy the worm-focused network intra-point of control, and achieve complete destruction of topology restoration points. The second-best goal is the same as the second-best goal in [Scenario 2].
  (4) Defender's Goal: Same as [Scenario 2].
  (5) Major attack sequence: Outside the topology → Attempt to break into the internal network → Temporarily break through UTM or Firewall terminal configured based on triplet → Win server 12 DNS Major invasion and Backup image neutralization and destruction → Achieve final occupation.
  (6) Major defense sequence: Carry out proactive deception and monitoring of hosts and terminals in the entire topology → Carry out additional monitoring based on multiplexing and host image integrity guarantee at set time intervals → Carry out attacker intrusion detection, reactive blocking-exit, and normal restoration to a random snapshot → Achieve complete exit to the outside.

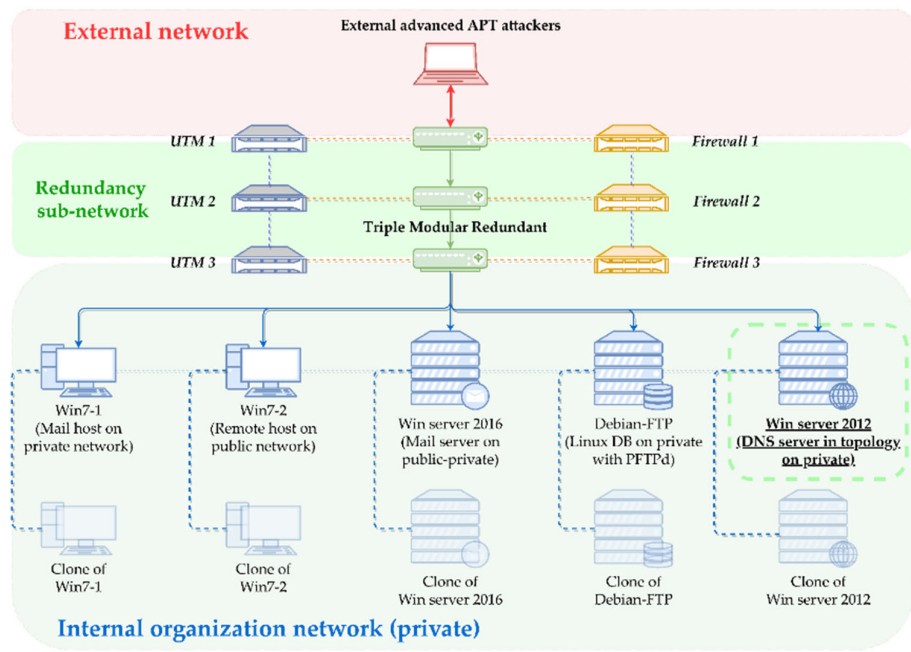

**Figure 6.** Overview and related topology with an organizational attack-defense strategy for Scenario 3.

MITER CVE vulnerability and CVSS scores constantized and calculated by scenario are structured as detailed in Table 3 and converted into a DB. Each vulnerability was selected for the ease of performing zero-sum-based requests and feedback of attackers and defenders based on limited views and surface knowledge possessed at any point in the CVE list that yields high CVSS focusing on Windows and Linux OS. These vulnerability scores are also applied as standard indexes when quantitative rewards by episode are subtracted or added. In addition, it is formulated in detail as a potential attack point based on weak services and protocols based on Ipv4 and socket ports for important hosts such as DNS servers to induce intrusion behaviors.

**Table 3.** CVE-based vulnerability table for deception with organizational defender or CKC-based APT attack with attacker.

| CVE ID | Vulnerability | Role in Org. | Related Node | CVSS 2.0 |
|---|---|---|---|---|
| CVE-2020-25223 | Remote code execution on web-admin in Sophos SG | IDS/IPS | UTM | 10.0 |
| CVE-2020-12271 | Remote code execution on web-admin in Sophos SFOS | Access control | Firewall | 7.5 |
| CVE-2017-0144 | *EternalBlue*. Remote code execution | Local host | Win7-1 | 9.3 |
| CVE-2019-0708 | *BlueKeep*. Use-after-free, Remote code execution | Remote host | Win7-2 | 10.0 |
| CVE-2020-1350 | *SigRed*. Heap overflow, Remote code execution | DNS server | Win server 2012 | 10.0 |
| CVE-2020-0796 | *SMBGhost*. Remote code execution | Mail server | Win server 2016 | 7.5 |
| CVE-2015-3306 | Remote command execution in ProFTPD 1.3.5 | PFTPD server | Debian-FTP server | 10.0 |

Both attack graph based on the CVE-based attack contact point and defense graph of the defender are conceptualized as adaptive mutations according to the network and host-based organizational OSINT with the same deception sequences as shown in Figure 7. In this case, directly mutating the unique authorized network band group used for normal service supply as direct mutation targets of MTD is unsuitable for all continuous organization service provision, channel migration, and communication operation schemes between internal network servers. Accordingly, the virtual network communication channel is additionally expanded based on the network fingerprint range in OSINT-based Decoy promised in advance at random mutation time intervals between internal users or certain permitted outsiders. MTD mutation targets are then selected. In addition, a rule table is formulated so that the public network information given to the host and the false virtual network information can communicate with each other in a

dynamic pair structure. The concept of judging whether or not the dynamic shifting of virtual network information should be performed according to changes in the external entropy such as the occupation of hosts following the attacker's success in partial invasion is also applied.

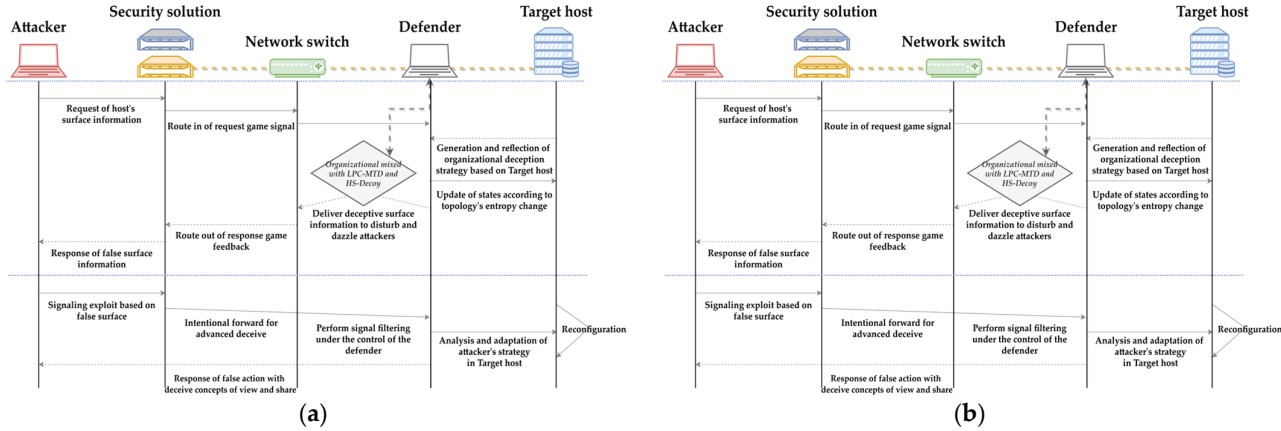

**Figure 7.** Examples of defender's episode-based deceptive sequences with MTD and Decoy concepts in SOD2G. (**a**) Mutation sequence based on network-layer elements. (**b**) Mutation sequence based on host-layer elements.

The deception sequence corresponding to Figure 7a is the reactive sequence related to MTD and decoy deceptions by host for cases in which access to the legitimate server is requested based on the network information group such as IP, port, network-layer service, etc. previously collected by the attacker. If the network specification in the packet requested by the attacker is not permitted at the present time point, false internal surface or vulnerability information is transmitted to the attacker through deceptive signaling to induce a cognitive judgment in the episode to be biased to the defender dominantly. In addition, to intentionally deceive the attacker to easily possess an isolated sandbox and drift, the defender artificially creates a false network information group corresponding to the information group requested by the attacker in the direction to minimize the attacker's suspicion, inserts it into the response packet, and transmits it. As a result, the deceived attacker gradually continues acting in the direction that maximize the defender's gain in a relevant episode and therefore loses the possibility to reach its final goal. Figure 7b is similar to, but different from Figure 7a in that it is a reactive deception sequence through which the attacker does not attempt attacks with IP or port-based network information, but instead executes attacks in units of application-layer service such as protocols (e.g., SSH and FTP), SMB and NetBIOS-based internal message sharing, RDP-based remote monitoring, and so on.

*4.2. Results*

This section details results of the MTD and Decoy-based attack-defense simulation experiments performed by scenario within the proposed SOD2G. A comparative analysis between different studies conducted by major metrics is also performed. Individual scenarios and attack-defense sequences are characterized with a focus on sub-party-based non-critical OSINT specifications as listed in Table A1. Overall parameters related to the experiment are also established based on Tables A2 and A3.

4.2.1. Comparative Results of Attacker-Defender Game Competition with Strategies

Figures 8–10 show result sets normalized by POMDP status in Figure 3 based on MTD and Decoy sampling and related signaling techniques customized based on existing MTD deception and distribution policies [21,35,46] for the degree of success of defense according to changes in the defender's final reward based on discount coefficients by

scenario, as well as organizational mix newly added with organizational OSINT. Figure 11 shows a set of results normalized for the degree of success of attacks according to the discount coefficient-based attacker's reward.

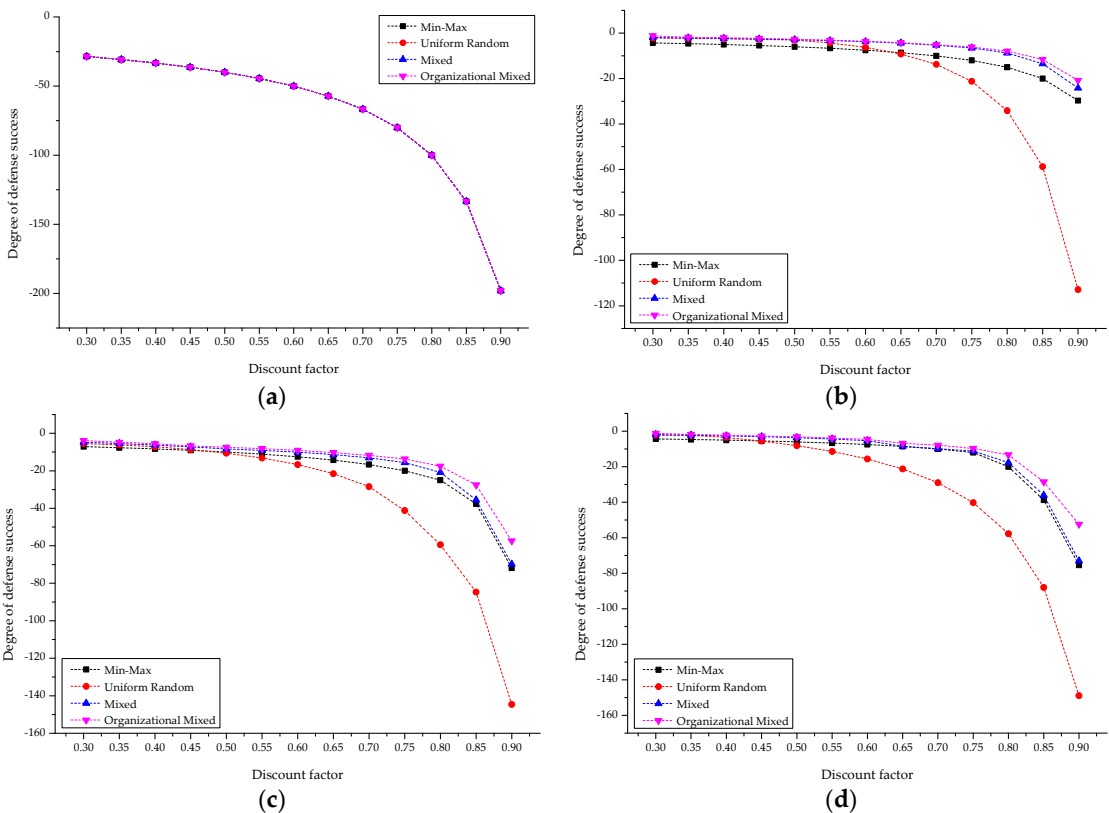

**Figure 8.** Comparison of the degree of defense success in each state with Scenario 1. (**a**) $S_0$, (**b**) $S_1$, (**c**) $S_2$, and (**d**) $S_3$.

Figure 8a shows the degree of defense success efficiency of the defender in Scenario 1 based on the $s_0$ state according to the discount factor that cuts off the range of judgment by player surfaced by episode to force them into limited microscopic views. Through the foregoing, to achieve the occupation of the final target invasion point or failure to reach as a result of the attacker finally being weeded out after reaching the maximum number of invasion attempts, the defender's total reward decreases exponentially as discount factor increases. The decreased total reward is always the same regardless which deceptive technique is additionally applied as a major deceptive strategy. In this case, the reward shows a tendency to linearly decrease until the discount factor reaches 0.7. It is calculated to be −29, −40, and −67 when discount factors are 0.3, 0.5, and 0.7, respectively. It decreases rapidly and exponentially as the discount factor increases beyond 0.7. Thus, extreme approximate values are derived, such as -198 when the discount factor is 0.9. That is, in a competitive environment, the lower the minimum discount factor value for the defender to force the attacker to make quick judgments by episode, thereby securing predominance, the more advantageous the discount factor value is to the defender. This indicates that the discount factor should be optimized not to exceed 0.7, even if it is increased by the attacker. This tendency proves that when the degree of time-based deceptive signaling, which has been calculated to actively attenuate the spatiotemporal allowance for the attacker to determine the optimum branch in the presence of the defender while minimizing the formulation of effective attack surface information, is higher and the time interval for mutation is shorter, the attacker's gain can be suppressed more and asymmetric inferiority can be forced further. This is also the case for Figure 8b–d, which are sets of partial comparison results for dynamic states other than $s_0$. Finally, if

the efficiency of the deceptive signaling scheme, which biases the attacker's early cognitive judgment to the defender's advantage, is improved as a single parameter, it is necessary to control the discount factor such that it does not exceed 0.7 for the position of the attacking player. Conversely, for the position of the deceptive defensive player, it is necessary to expand the market share in the competitive environment based on false signaling and feedback to increase the discount factor above 0.7.

Figure 8b–d with the same discount factor metric show defense success efficiency of the defender in Scenario 1 based on states $s_1$, $s_2$, and $s_3$, respectively.

First, in Figure 8b, in the $s_1$ state related to a single UTM deployment with improved network monitoring and access control functions, it was revealed that even if the discount factor that gave the attacker a high asymmetric predominance increases, defense success efficiency values of deception techniques except for Uniform Random did not decrease much. Discount factor values converged from 0.9 to between −20 and −30. In this case, Mixed derived an improvement in defense efficiency by 126% at the maximum and 21% at the minimum compared to Min-Max, which was the performance measure baseline. Organizational Mixed also produced improvements in defense efficiency by about 58% at the maximum and 16% at the minimum compared to Mixed. This tendency proves that since Uniform Random, which determines a single host as the target of concentration of the defender's deceptive signaling and monitoring and reactive response with a uniform probability distribution, distributes invasion probability to upper entities that transmit unique and high ripple effects such as the UTM in Scenario 1 in an unconditionally equal manner, even for those that do not execute attacks, the attacker cannot judge that it can carry out an invasion in earnest only after infiltrating and breaking in through the defense solution. In addition, since Mixed and Organizational Mixed represent mixtures of Min-Max and Uniform Random and thereafter apply and strategize different probability distributions to different hosts according to degrees of changes in entropy such as the attacker's invasion preference, observable-determinable topology state, and the attacker's environment, they have further improved the deception defendability. It was also proved that Organizational Mixed derived an additional performance improvement compared to Mixed since it added more subdivided deceptive elements based on network, host, and service layers. It additionally secured substantiality as OSINT. This was similar in Scenario 2 of Figure 9 and Scenario 3 of Figure 10, except for the special tendency of Uniform Random. Finally, if it is intended to minimize the decreased efficiency of the deceptive defense for a single security solution between the internal network and the external network, the share should be ensured such that the defender's inferior discount factor is also configured to be more than 0.75 but lower than 0.8 without selecting Uniform Random.

Next, in (c), although Uniform Random showed the most inefficient defense success degree even in the $s_2$ state related to the achievement of a weak invasion through the early search in the topology and the first exploit, unlike before, it was revealed that the defense success efficiency rapidly decreased from when the discount factor reached 0.8 to converge between −55 and −75. In this case, Mixed derived an improvement in defense efficiency by 48% at the maximum and by 3% at the minimum compared to Min-Max. Organizational Mixed also resulted in improvements in defense efficiency by 22% at the maximum and 10% at the minimum compared to Mixed. This tendency proves that the efficiency of Uniform Random is reduced drastically due to the unique high ripple effect based on the internal communication channels and shared directories installed by the host on the Linux DB server, which is the final invasion target point of the attacker, along with the fact that the degree of specificity is high for both improved Mixed and Organizational Mixed. This is also the case in Scenario 2 of Figure 9 and Scenario 3 of Figure 10. Therefore, if it is intended to minimize the possibility of success of internal reconnaissance and early exploitation by the APT attacker who is performing initial surfacing to improve attack efficiency after the initial intrusion into the organization's internal network, the attacker

should be perturbed such that Uniform Random is not selected and that the attacker dominant discount factor is configured to be lower than 0.75.

Finally, (d), that is, the $s_3$ state related to the achievement of the final invasion goal after the lateral movement and serial chain advancement following the initial exploit, has almost the same pattern as (c). Mixed showed an improvement of defense efficiency by 121% at the maximum and 3% at the minimum compared to Min-Max. Organizational Mixed also yielded improvements of defense efficiency by 62% at the maximum and 14% at the minimum compared to Mixed. Finally, if it is intended to highly prevent and actively block the achievement of the final invasion by the attacker who performs lateral movement and multiple rough privilege elevation attacks based on the contact point after the success of initial occupation in the organization's internal network, it is judged that a large number of deceptive signals should be rapidly projected and made into noise without selecting Uniform Random, similarly to (c), so that the attacker dominant discount factor is configured to be equal to or higher than 0.75 but lower than 0.8.

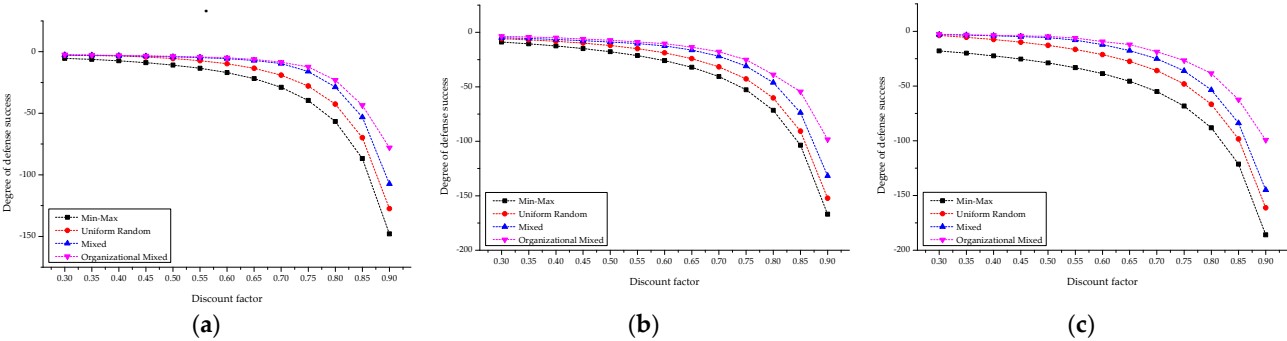

**Figure 9.** Comparison of the degree of defense success in each state with Scenario 2. (**a**) $S_1$, (**b**) $S_2$, and (**c**) $S_3$.

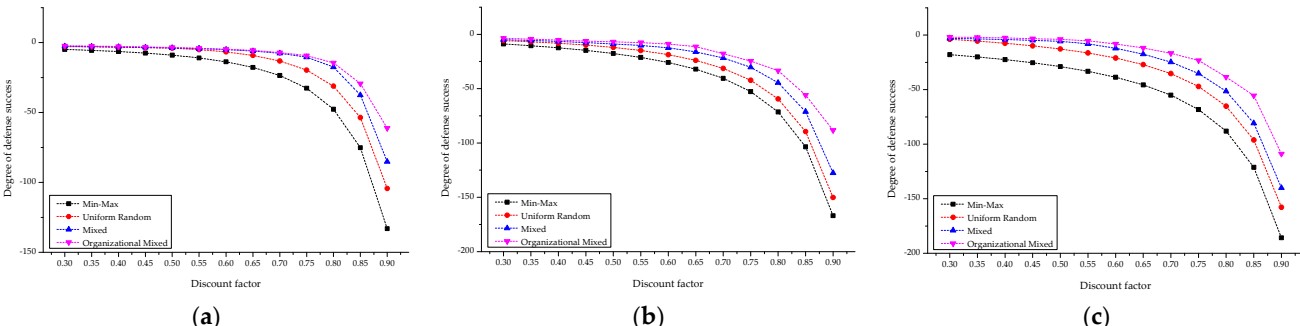

**Figure 10.** Comparison of the degree of defense success in each state with Scenario 3. (**a**) $S_1$, (**b**) $S_2$, and (**c**) $S_3$.

Since Figures 9 and 10, respectively, simulated competition based on Scenario 2 and Scenario 3, wherein concepts of multiple security solutions, hosts, and final invasion target points, to which a single ripple effect was distributed, were defined differently, a tendency for Min-Max (not Uniform Random) to show the lowest defense efficiency was commonly derived, unlike in Figure 8. The superiority relationship of the defense efficiency of other deception techniques was the same as that shown in Figure 8.

Figure 9a shows the degree of defense success efficiency of the defender in Scenario 2 based on the $s_1$ state. In this case, after the discount factor reached 0.7, the defense success efficiency levels of all deception techniques decreased sharply and drastically to converge on −75 to −150 at 0.9. In addition, Mixed showed improvements in defense efficiency of 101% at the maximum and 4% at the minimum compared to Uniform Random. Organizational Mixed also yielded improvements in defense efficiency by 38% at the maximum and 7% at the minimum compared to Mixed. This tendency was

attributable to the fact that as the single UTM solution in the previous scenario 1 was horizontally multiplexed as multiple UTM and firewall solutions, the ripple effect concentrated on lower-level hosts was dispersed such that a uniform distribution of invasion probability of Uniform Random was feasible. This was also proved by the fact that the final point of invasion was changed to a DNS server host that probabilistically distributed the transition ripple effects by host through mutual communication with security solutions other than the Linux DB server and an external publicly approved DNS service. Through the foregoing, it can be seen that to minimize the attenuation of defensive deception efficiency for vertically-multiplexed security solutions, the real share in the topology should be secured such that the attacker dominant discount factor is configured to be lower than 0.7 without selecting the Min-Max.

Next, (b) shows defense success efficiency in Scenario 2 based on the $s_2$ state. Mixed improved defense efficiency by about 51% at the maximum and 14% at the minimum compared to Uniform Random, whereas Organizational Mixed improved defense efficiency by about 36% at the maximum and 23% at the minimum compared to Mixed. Finally, the method to actively reduce the possibility of attack success in the attacker's first intrusion and initial exploitation attempt in the organizational topology of the DNS server host should be configured similarly to (a) such that the discount factor is maintained between 0.7 and 0.75.

(c) also shows defense success efficiency in the $s_3$ state based on Scenario 2. It was judged that related countermeasures could be formulated, such as configuring the discount factor to be lower than 0.65 in an approach similar to (a) and (b).

Results of SOD2G experiments based on Scenario 3 related to the operation strategies of multiple UTM and firewall solutions triplicated based on vertical redundancy, horizontal diversity, and hosts considering restorability and resilience by cloning candidate images based on snapshots are compared and analyzed. Result sets are shown in Figure 10.

Figure 10a shows defense efficiency in the $s_1$ state related to the deployment and operation of triplicated UTM and firewall solutions. In this case, Organizational Mixed improved defense efficiency by 39% at the maximum and 7% at the minimum compared to Mixed and converged as a sharp negative gradient when the discount factor was in the range of 0.75 to 0.8, whereby it was finally derived to be between −60 and −135. The tendency as such proved that the triplicated security operation environment was selected due to the fact that when an invasion had occurred, the defender's risk in this environment was lower than in other environments while the attacker's high robustness for solution search and breakthrough could be ensured.

Figure 10b shows the degree of defense efficiency in the $s_2$ state related to the attacker's initial reconnaissance and exploitation attempts on hosts inside the organization that have cloned snapshot images. Organizational Mixed achieved improvements in defense efficiency by 44% at the maximum and 21% at the minimum compared to Mixed. Defense efficiency sharply decreased when the discount factor was in the range from 0.75 to 0.8, finally converging to be between -90 and -170. This tendency proves that even if the restorability and resilience level by host are improved with redundancy-based additional snapshots, if invasion and occupations are performed within the length of the time interval for snapshot configuration, the environment will revert to an environment similar to the defender intelligence already possessed by the attacker such that the defender information validity remains preserved.

Figure 10c shows the degree of defense efficiency in the $s_3$ state related to the attacker's final invasion and achievement of the target point. Organizational Mixed derived improvement in defense efficiency by 59% at the maximum and 28% at the minimum compared to Mixed. Defense efficiency sharply dropped when the discount factor was in the range from 0.65 to 0.7, finally converging to be between −110 and −190. Through the foregoing, it is judged that internal deceptive signaling to secure a discount factor lower than 0.6 should precede in order to prevent an attacker from gaining

predominance and adding rewards in a closed network in which security has finally been secured based on redundancy and diversity.

Figure 11 shows that the attack success efficiency improvement is derived based on the positive gradient as the attacker dominant discount factor increases based on the surfaced defender information through search and reconnaissance in Scenario 3. Combined with the fact that the sum of the attacker's total reward and the defender's total reward is always zero, it can therefore be concluded that attacks and defenses are always performed on a zero-sum basis.

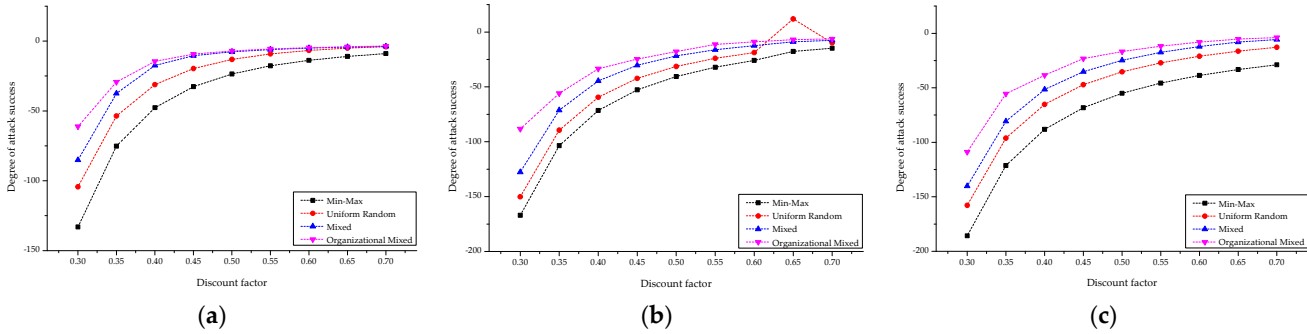

**Figure 11.** Comparison of the degree of attack success in each state with Scenario 3. (**a**) $S_1$, (**b**) $S_2$, and (**c**) $S_3$.

#### 4.2.2. Sensitivity Analysis by Deceptive Metrics

Next, sensitivity analyses were carried out in detail using major metrics for the concepts of LPC-MTD and HS-Decoy-based social engineering-type organizational deceptive defense. Figure 12 shows sets of analysis results regarding the possibility of success of deceptive signaling based on the increase in surface uncertainty and limited effectiveness following the defender's active deception normalized by scenario. Figures 13 and 14 show sets of results obtained by reclassifying metrics in Figure 12 into MTD and Decoy-based detailed metrics and normalizing them.

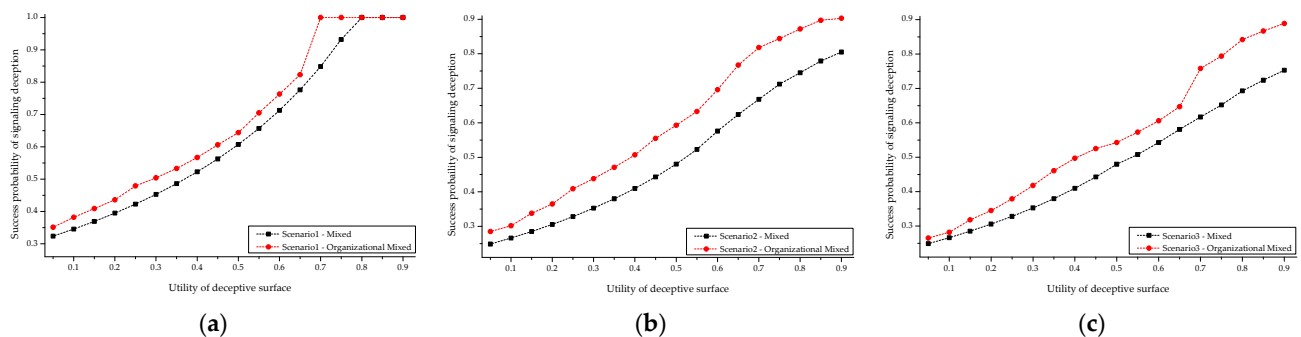

**Figure 12.** Comparative analysis of the overall success probability of signaling deception in SOD2G with each scenario. (**a**) Scenario 1, (**b**) Scenario 2, and (**c**) Scenario 3.

First, possibilities of success of deceptive signal projection in Figure 12a–c are probabilistic comparison metrics that strategize the active deception events that deceive the occupancy and surface by disrupting the attacker's perception related to the invasion target point in the defender's favor, or by inducing the attacker to a false host occupation environment with detailed deception elements such as MTD mutation-based disinformation and artificial exposure, Decoy-based cloning and mimicking, and OSINT information group, which are associated with the proactive defense and signaling component in Figure 2. These detailed elements are composed of detailed strategies in Organizational Mixed. Based on the foregoing, it can be seen that compared to the existing

Mixed, Organizational Mixed achieved at least 23% higher deception efficiency by increasing the uncertainty of the surface information by host and reducing the effectiveness. It can achieve the stage at which it completely deceives the attacker earlier at the deception surfacing value of 0.7. This shows that since the existing Mixed simply focuses on selecting a transition path that maximizes the reward to be obtained in the next episode, and since the OSINT-based disinformation, artificial exposure concept, social engineering decoy, etc. to obtain the highest final reward even if some loss should be born have yet to be composed, and the relevant detailed deception has not been clearly defined in the presence of the defender, Organizational Mixed considering all of them exhibits higher deception efficiency. This also proves that when establishing deception policies by host, policies should not be established for naïve application to entire entities. They should be established for application to entities characterized as detailed entities layered with IPv4 and port, socket, network service, application services, and so on. Through these characteristics, it was finally revealed that Organizational Mixed had higher deception performance in SOD2G than the existing Mixed.

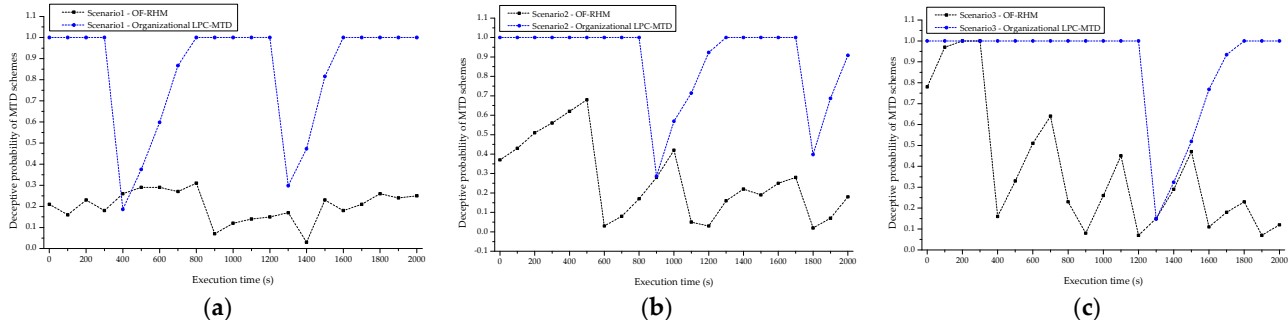

**Figure 13.** Comparative analysis of deceptive efficiency of disinformation and disclosure-based loosely mutations. (**a**) Scenario 1, (**b**) Scenario 2, and (**c**) Scenario 3.

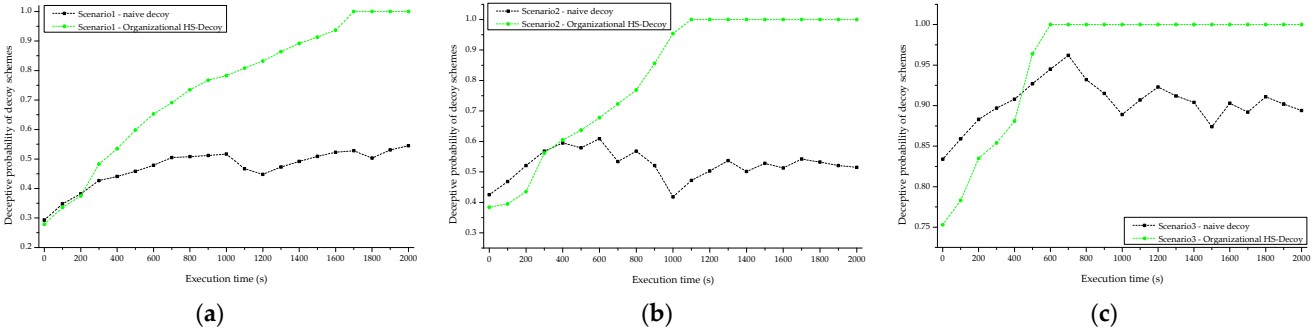

**Figure 14.** Comparative analysis of deceptive efficiency of cloning and mimicking-based hierarchical OSINT decoys. (**a**) Scenario 1, (**b**) Scenario 2, and (**c**) Scenario 3.

The possibility of MTD-based deception in Figure 13a–c of is a probabilistic comparison metric wherein the degree to which the attacker's attack attempts are induced is atomically normalized so that the final invasion target host in the topology surfaced based on CVE vulnerability or decoy beacon in an episode where the LPC-MTD or OF-RHM-based existing MTD scheme is implemented toward defender dominance is isolated from the actual host with false information surfaced based on OSINTs by organization. This comparison metric is also associated with the proactive defense and signaling behavior component in Figure 2. Based on these correlation metrics, it can be seen that Organizational LPC-MTD has a five times higher possibility of deception through disinformation and artificial information disclosure on average compared to the existing OF-RHM as well as at least three times higher signaling-based intentional exposure rate

on false defender surfaces involving OSINT on average. This demonstrates that, unlike the existing MTD, LPC-MTD reactively lowers the mutation strength according to the analyzed attacker's attack routine to deceive the attacker by forcing a judgment such that it can successfully bypass the MTD along with the fact that the related reliability can be improved with disinformation and artificial information exposure based on the deceptive OSINT layered referring to the target host. That is, it can be also identified that the OF-RHM scheme is constructed to only have deception possibilities not exceeding 31% for Scenario 1, 68% for Scenario 2, and 97% for Scenario 3. However, Organizational LPC-MTD is operated in order to deceive the attacker while always maintaining 100% convergence. When the attacker carries out invasion, it is quickly reconstructed with a separate deceptive surface that is not much related to the previous surface and restored to converge on 100%. The foregoing finally shows that LPC-MTD has a higher deception possibility in SOD2G than existing OF-RHM.

Finally, the possibility of decoy-based deception in Figure 14a–c is represented by probabilistic comparison metrics obtained by normalizing the degree to which the attacker can be induced and thereafter maximally isolated under the intention of the defender according to true-false occupancy rates by player along with the concept of a limited view in an episode where HS-Decoy and the existing naive Decoy scheme are created, distributed, and managed. This comparison metric is associated with the proactive defense, signaling, and reactive defense components shown in Figure 2. Based on the foregoing, it can be identified that in the case of Organizational HS-Decoy, the possibility of deception through OSINT-based cloning and mimicking is 37% higher on average compared to the existing naive Decoy and the degree of improvement of decoy quality utilizing the actual characteristics of any legitimate host within the topology is at least 163% higher on average. A positive gradient is always maintained. This is based on the fact that unlike naive Decoy, in the case of HS-Decoy, the number of deceptive entities is large and the structure is stratified based on network, host, and service such that the degrees of redundancy and diversity are enhanced. The reliability is also enhanced based on OSINTs by organization. This can be confirmed through results in the condition wherein deception efficiency gaps according to the execution times by scenario linearly increase. This ultimately shows that HS-Decoy has a higher deception possibility in SOD2G than existing naive Decoy.

### 4.2.3. Optimized Values with Organizational Deceptive Game Models

Through the aforementioned Sections 4.2.1 and 4.2.2, overall optimized values for maximizing the defensive deception efficiency of the defender by game, deception metric, and related parameter in the SOD2G framework as well as minimizing the asymmetric advantage of the attacker are simplified as detailed in Table 4.

**Table 4.** Optimized defender values with game and deceptive metrics and parameters in SOD2G framework.

| Metric and Related Parameter | Targeted Scenario | State | Value |
|---|---|---|---|
| Attacker's discount factor | Scenario 1 | $S_0$ | <0.7 |
| | | $S_1$ | <0.75~0.8 |
| | | $S_2$ | <0.7~0.75 |
| | | $S_3$ | <0.75~0.8 |
| | Scenario 2 | $S_1$ | <0.7 |
| | | $S_2$ | <0.7~0.75 |
| | | $S_3$ | <0.65 |
| | Scenario 3 | $S_1$ | <0.75 |
| | | $S_2$ | <0.7 |
| | | $S_3$ | <0.6 |
| Utility of deceptive surface | Scenario 1 | $S_3$ | >0.6 |
| | Scenario 2 | $S_3$ | >0.7 |

| Metric and Related Parameter | Targeted Scenario | State | Value |
|---|---|---|---|
| | Scenario 3 | $S_3$ | >0.75 |
| | Scenario 1 | $S_3$ | >0.2 |
| Minimum mutation strength for organizational LPC-MTD | Scenario 2 | $S_3$ | >0.3 |
| | Scenario 3 | $S_3$ | >0.15 |
| | Scenario 1 | $S_3$ | >0.75 |
| Minimum decoying signal strength for organizational HS-Decoy | Scenario 2 | $S_3$ | >0.85 |
| | Scenario 3 | $S_3$ | >1 |

## 5. Discussion

In this study, PBNE, BSSG, and POMDP-based SOD2Gs were proposed to improve the practical operability of defensive cyber deception and optimize defender's deception strategies against attackers by organization. In addition, the attack-defense decision-making process of mixed LPC-MTD which selects the MTD mutation strength with the advantage of the defender and HS-Decoy, which applies a deception concept with the goal of deceiving the attacker based on OSINT, are diversified by scenario. Through the foregoing, deception attack-defense efficiency levels optimized by organization could be calculated based on game equilibrium. Independent defensive deception values necessary to minimize operational performance degradation and maximize security without additional procedures or separate application of dedicated protocols could also be formulated by scenario.

First, when the defense success efficiency against the discount factor for the attacker's advantage was composed as a main comparison index and summarized, the organizational mixed showed an average performance improvement of 40% or more compared to the mixed by scenario. To minimize attacker dominance in the topology, the attacker discount factor should also be suppressed so as not to exceed 0.7 on average. Next, in the sensitivity analysis for optimizing game models in SOD2G, LPC-MTD, and HS-Decoy based on OSINT yielded improvements of at least 60% and 30% in deception efficiency, respectively, compared to the existing model. Optimal values by metric and by parameter were also produced by empirical calculations through comparative analysis.

However, the following improvement measures are needed to address limitations of this study.

- Reliability and Substantial Issues: To prevent both potential legal disputes and research ethics problems, characteristics by organization intellectualized in the game should be pre-processed and extracted by element within the limited range in the third party-based public OSINT information group. However, this will soon be different from the unique policies and principles related to the actual organization, leading to clear practical differences from the organizational environment where open and closed networks are operated separately. In addition, even when OSINTs are collected and formulated, since individual organizations have different numbers of elements, vertical-horizontal relationships, ripple effects, and vulnerability, additional augmentation and normalization should be performed.
- Scalability and operability issues: Since the proposed study conducted experiments focusing on minimizing the degradation of availability in the defender environment and maximizing the defender predominance based on disinformation and artificial information exposure, scalability issues might occur in scenarios or domains that are not considered. Therefore, rather than concretizing the attack-defense probability value at the scenario level, the probability value should be abstracted at the organizational domain classification level and the evaluation verification process should be newly constructed.
- Issue of additional demand for strategy based on hyper-game: In an actual attack-defense environment, players can subjectively process and determine asymmetric dominance or subordinance information that is accessible at the present time point.

This proves that under the premise that players have consistent views and that they cannot be represented solely by a naive game theory that models dynamic decision-making in an uncertain situation. This study tried to alleviate some conflicts of views between players by conceptualizing the disinformation-based partial signaling game tactic and dynamically changing the state-transition probability according to changes in the attack-defense state based on the deceptive signal. However, related potential side effects will still remain. Accordingly, considering recently studied hyper games [33], it is necessary to quantify decision-making when selecting the best strategy by considering all players' subjective or wrong beliefs and their perceived uncertainty at any point of time. A strategy set selection calculation method that processes a large number of solution spaces required for modeling and obtains an optimal solution should also be configured separately.

## 6. Conclusions

In this study, a PBNE and BSSG-based zero-sum game foreground and POMDP state-transition background-based SOD2G characterized as organizational OSINTs were proposed. To this end, the decision-making process was non-sequentially stratified based on an incomplete and deceivable organizational environment and vulnerability information, such that it could construct proactive deception considering the organizational domain and simulate the related deception efficiency improvement. Defender deception strategies to achieve both high reactive response rates and low false-positive and false negative rates could also be calculated to converge on optimal values by scenario. In addition, to improve the practicality and operability related to the customized deception concept, it was possible to quantify the organizational game model based on the scenario. In future studies, to secure the empirical reliability of deception efficiency levels of the proposed SOD2G and the detailed model, and to formulate sophisticated cyber threat intelligences by organization, attack-defense scenarios should be constructed based on operational behaviors to diversify all related deceptive countermeasures.

**Author Contributions:** Conceptualization, S.S.; methodology, S.S.; software, S.S.; validation, S.S. and D.K.; formal analysis, S.S. and D.K.; investigation, S.S.; resources, S.S.; data curation, S.S.; writing—original draft preparation, S.S. and D.K.; writing—review and editing, S.S. and D.K.; visualization, S.S.; supervision, D.K.; project administration, S.S. and D.K; funding acquisition, D.K. All authors have read and agreed to the published version of the manuscript.

**Funding:** This research was funded by a grant (No. NRF-2020R1F1A1068774) of the National Research Foundation of Korea (NRF) funded by the Korea government (MEST).

**Acknowledgments:** This research was funded by a grant (No. NRF-2020R1F1A1068774) of the National Research Foundation of Korea (NRF) funded by the Korea government (MEST) and also supported by Kyonggi University's Graduate Research Assistantship 2021.

**Conflicts of Interest:** The authors declare no conflict of interest.

## Appendix A. Supplementary Data

**Table A1.** Overview of organizational deception knowledge with OSINT for MTD and decoy-based game competitions.

| Element of Deception Knowledge | Description and Related Examples |
| --- | --- |
| Network topology and infrastructure | Public or private network infrastructure<br>ex) firewall-router-DMZ-NAC-host-service |
| Network device | Device for internal and external communication in organizational infrastructure ex) CISCO ISR router, CISCO catalyst switch |
| Security solution | Solution for detecting, blocking and deceiving attackers in topology and hosts ex) Sophos XG firewall, Sophos SG UTM, Splunk SOAR (security orchestration,   automation and response) |

| Element of Deception Knowledge | Description and Related Examples |
|---|---|
| IPv4 address | Public or private IPv4 with prifix and subnet rule<br>ex) 11.19.0.0/16, 137.0.0.0/13, 205.79.192.0/19 |
| Port | Public socket for inbound and outbound communication with network service<br>ex) well-known, registered, and dynamic on 0~65535 |
| OS | Operating system fingerprint by each host<br>ex) Windows 7 SP1, Windows Server 2012 and 2016, Debian, Rocky Linux |
| Service | Protocol-based services in network and system-layer<br>ex) FTP, SSH, DNS, NetBIOS, SMTP, RPC, HTTP/HTTPS |
| Vulnerability | CVE and CVSS 2.0-based vulnerabilities in topology and hosts<br>ex) CVE-2017-0144, CVE-2020-1350, CVE-2015-3306 |

**Table A2.** Examples of deceptive MTD and decoy-based major parameters with defender model in SOD2G.

| MTD Parameter | Value | Decoy Parameter | Value |
|---|---|---|---|
| Time slot length for periodic mutation (s) | 1–100,000 | Activation time (s) | 0–259,200 |
| Mutation batch pool size | 64–2048 | Number of decoying hosts | 0–10 |
| Bellman-based mutation sampling size | 8–2048 | Number of decoying services | 0–40 |
| Number of surface views in topology | 0–100 | Number of decoying vulnerabilities | 0–20 |
| Maximum number of branches in attack graph | 10 | Number of decoying beacons | 0–50 |
| Maximum number of deceptive signaling | 50–200 | Number of decoying signals | 1–10 |
| Mutation range of security solutions | 0–6 | Level of vulnerabilities | L-M-H |
| Mutation range of IPv4 addresses | $2^8$–$2^{36}$ | Maximum number of compromised decoys | 0–3 |
| Mutation range of port numbers | $2^{10}$–$2^{16}$ | Maximum number of defender sandboxes | 0–1 |
| Mutation range of OS fingerprints | 0–10 | Degree of OSINT-based cloning scheme | 0.01–0.90 |
| Mutation range of services | 0–20 | Degree of OSINT-based mimicking scheme | 0.01–0.80 |
| Mutation range of vulnerabilities | 0–10 | Degree of enticingness with decoy | 0.30–1.00 |
| Degree of OSINT-based disinformation | 0.01–1.00 | Degree of conspicuousness with decoy | 0.50–1.00 |
| Degree of OSINT-based artificial disclosure | 0.01–1.00 | Degree of variability with decoy | 0.10–1.00 |
| Degree of reliability of MTD-based signal | 0.01–1.00 | Degree of differentiability with decoy | 0.70–1.00 |

**Table A3.** Configuration of experimental game parameter for decision making each episode in SOD2G.

| Parameter | Value |
|---|---|
| Simulation time (s) | 3600–259,200 |
| Number of simulation run | 1–10 |
| Attack time (s) | 1800–129,600 |
| Defense time (s) | 1800–129,600 |
| Number of scenarios | 3 |
| Number of CKC phases with attacker | 4–7 |
| Maximum number of attacker compromise attempts | 1–10 |
| Operating system | Ubuntu 20.04 LTS |
| Language | Python 3.9.2 (Anaconda) |
| Game solver | Gurobi optimizer 9.0, IBM CPLEX, Google Or-tools |

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
