# Peer review of "SOD2G: A Study on a Social-Engineering Organizational Defensive Deception Game Framework through Optimization of Spatiotemporal MTD and Decoy Conflict"

_electronics, doi:10.3390/electronics10233012_

Round 1

Reviewer 1 Report

Manuscript proposes a deception concepts and game models for handling
incomplete private information in organizational environments. The English needs significant improvement as the manuscript is very difficult to follow.

Abstract is too long, yet its not informative. Results and achievement of the study must be communicated in the abstract. 

Although the paper has appropriate length and informative content, several parts must be improved and written in better grammar and syntax. It would be essential if authors would consider revising the organization and composition of the manuscript, in terms of the definition/justification of the objectives, description of the method, the accomplishment of the objective, and results. The paper is generally difficult to follow.

Paragraphs and sentences are not well connected. Furthermore, I advise considering using standard keywords to better present the research.

Use better keywords; the standard keywords must be used. Please revise the abstract according to the journal guideline. It must be under 200 words. The research question, method, and the results must be briefly communicated. The abstract must be more informative. I suggest having four paragraphs in the introduction for; describing the concept, research gap, contribution, and the organization of the paper. The motivation has the potential to be more elaborated. You may add materials on why doing this research is essential, and what this article would add to the current knowledge, etc. The originality of the paper is not discussed well. The research question must be clearly given in the introduction, in addition to some words on the testable hypothesis. Please elaborate on the importance of this work. Please discuss if the paper suitable for broad international interest and applications or better suited for the local application? Elaborate and discuss this in the introduction.

How the game theory and other the entire model had been validated. Please also elaborate on the data. 

State of the art needs improvement. A detailed description of the cited references is essential. Several recently published papers are not included in the review section. In fact, the acknowledgment of the past related work by others, in the reference list, is not sufficient, some relevant references to game theory can be read to better point out the reliability and power of game theroty e.g., Determining the contribution of environmental factors in controlling dust pollution during cold and warm months of western Iran using different data mining algorithms and game theory. Consequently, the contribution of the paper is not clear. Furthermore, consider elaborating on the suitability of the paper and relevance to the journal. Kindly note that references cited must be up to date.    

Author Response

I would appreciate it if you could refer to the attached "Rebutal_comments" document.

Reviewer 2 Report

Comments for Authors 

  • What is the motivation of the proposed work?
  • Introduction needs to explain the main contributions of the work clearer.
  • The novelty of this paper is not clear. The difference between present work and previous Works should be highlighted.
  • Authors must explain in detail the introduction section.
  • Authors must develop the framework/architecture of the proposed methods
  • There is need of flowchart and pseudocode of the proposed techniques
  • Proposed methods should be compared with the state-of-the-art existing techniques
  • Research gaps, objectives of the proposed work should be clearly justified.
  • To improve the Related Work and Introduction sections authors are highly recommended to consider these high quality research works <On the Security and Privacy Challenges of Virtual Assistants>, <Cost-Efficient Service Selection and Execution and Blockchain-Enabled Serverless Network for Internet of Medical Things>
  • English must be revised throughout the manuscript.
  • Limitations and Highlights of the proposed methods must be addressed properly
  • Experimental results are not convincing, so authors must give more results to justify their proposal.

Major changes are required 

Author Response

(The authors gave the same response as above.)

Round 2

Reviewer 2 Report

Authors have improved the manuscript, so can be accepted